# AlignVid: Taming Visual Dominance via Training-Free Attention Modulation in Text-guided Image-to-Video Generation

Yexin Liu[1]  Wenjie Shu[2]  Zile Huang[3]  Haoze Zheng[1]  Yueze Wang[4]
Manyuan Zhang[5]  Jinjing Zhu[6]  Sernam Lim[3]  Harry Yang[1]

## Abstract

Text-guided image-to-video generation has made substantial progress, yet it still struggles to execute text-specified edits that require substantial changes to a reference image (*e.g., object addition, removal, or modification*). Empirically, our analysis reveals that this stems from **visual dominance**, where the reference image causes severe attention dispersion, inhibiting the model's ability to incorporate new semantic information. To address this, we propose **AlignVid**, a training-free intervention that recalibrates the model's internal attention distribution. Drawing on an energy-based perspective of attention, AlignVid employs Attention Scaling Modulation (ASM) to reduce attention entropy and concentrate focus on semantic tokens, alongside Guidance Scheduling (GS) to maintain generation stability. To rigorously assess this capability, we present **OmitI2V**, a comprehensive benchmark for evaluating prompt adherence across object modification, addition, and deletion. Extensive experiments demonstrate that Align-Vid effectively enhances semantic fidelity with negligible computational overhead. Code and the OmitI2V benchmark are available at `https://github.com/LAW1223/AlignVid`.

## 1. Introduction

Image-to-video (I2V) generation aims to generate a temporally coherent video from an image. Early I2V methods predominantly focused on motion extrapolation (Blattmann et al., 2023; Wang et al., 2023a; Xing et al., 2024; Chen et al., 2023; Zhang et al., 2023; Zeng et al., 2024). Text-guided image-to-video (TI2V) extends this setting by conditioning the generative process on text prompts alongside the reference image, enabling fine-grained control over motion semantics and temporal dynamics (Kong et al., 2025; Wan et al., 2025; Chen et al., 2025; Zhang & Agrawala, 2025; Xu et al., 2024). However, current TI2V methods still fail to adhere to fine-grained prompt semantics, particularly when prompts prescribe substantial transformations of the reference image (e.g., adding, deleting, or modifying objects). As illustrated in Figure 1, given the prompt *"A sunflower grows in front of the house"*, the generated video preserves the image without inserting the sunflower, indicating a misalignment between the prompt and the generated video.

To investigate this, we conduct a pilot study which reveals a counter-intuitive finding: introducing a slight Gaussian blur to the input image improves both semantic fidelity and motion degree (Figure 2). Specifically, under the standard unblurred setting, the reference image induces strong **visual dominance** over the model's attention mechanism—this suppresses the text modality and causes severe attention dispersion (characterized by high entropy) in the video modality, a critical imbalance that directly impairs the model's ability to incorporate new semantic information and inhibits motion generation. In contrast, blurring the reference image alleviates such visual dominance, which in turn leads to a relative rise in text attention weights and a reduction in video attention entropy (Figure 2 (c)). This attentional recalibration restores the competitive priority of the text prompt, enabling it to effectively guide the video's semantic transformation as specified by the edit instructions. However, as input-level degradation inevitably compromises aesthetic quality, we pose a research question: *Can we directly regulate the model's internal attention distribution—mimicking this entropy-reduction effect without altering the input—to enhance semantic alignment while keeping its impact on visual quality (e.g., aesthetic) controllable?*

To this end, we revisit pretrained I2V models' attention mechanism via an energy-based lens, enabling precise attention entropy modulation to alleviate the identified vi-

---

[1]Hong Kong University of Science and Technology, Hong Kong SAR, China [2]ZODA, HangZhou, China [3]University of Central Florida, Orlando, FL, USA [4]Beijing Academy of Artificial Intelligence, Beijing, China [5]The Chinese University of Hong Kong, Hong Kong SAR, China [6]Hong Kong University of Science and Technology (Guangzhou), Guangzhou, China. Correspondence to: Harry Yang <harryyang.hk@gmail.com>.

*Proceedings of the 43rd International Conference on Machine Learning*, Seoul, South Korea. PMLR 306, 2026. Copyright 2026 by the author(s).

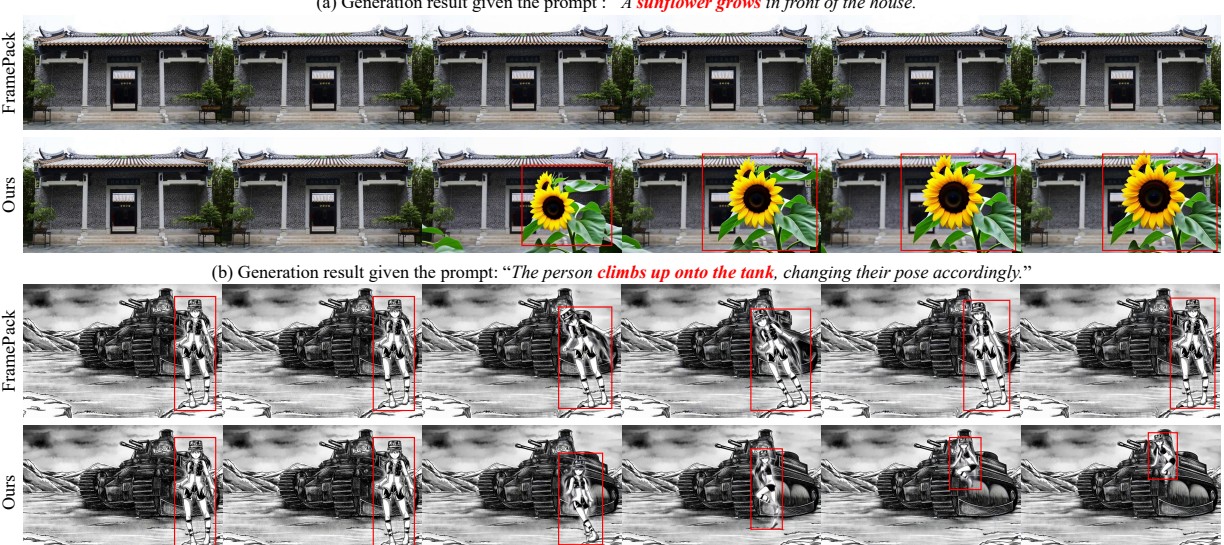

*Figure 1.* The baseline model (FramePack) exhibits *semantic negligence*, failing to realize the prompt-specified modifications. In (a), the sunflower mentioned in the prompt is entirely missing. In (b), the person remains static instead of climbing onto the tank as instructed.

sual dominance. Prior work (Hong, 2024) establishes attention as a gradient step minimizing an implicit energy function, laying a theoretical foundation for reshaping attention distributions. Complementing this, our observation shows pretrained models inherently achieve coarse foreground–background separation—distinguishing text-targeted semantic regions from background visual priors (the root of visual dominance). Building on this, we propose **AlignVid**, a training-free method for improving semantic fidelity through minimal intervention. Specifically, AlignVid comprises two components: (i) *Attention Scaling Modulation (ASM)*, which rescales query or key representations to **sharpen the energy landscape**. This recalibration yields a more concentrated, lower-entropy attention distribution that amplifies the prominence of text tokens over visual priors. (ii) *Guidance Scheduling (GS)*, which activates ASM selectively across transformer blocks and denoising steps to stabilize generation and mitigate visual-quality degradation. Unlike input-level perturbations such as blurring, which *visibly corrupt* the reference image, AlignVid performs this attention reallocation entirely within the model, thereby providing a tunable semantic–quality trade-off without input-level corruption. To evaluate semantic fidelity, we introduce **OmitI2V** benchmark. It comprises 367 human-annotated samples across modification, addition, and deletion.

Our main contributions can be summarized as: (i) **Problem Analysis.** We formalize *semantic fidelity challenge* in TI2V and empirically link it to *visual dominance*. Through a pilot study, we demonstrate that attention concentration (lower entropy) is essential for liberating textual signals from visual redundancy. (ii) **Method-AlignVid.** We propose a training-free framework that modulates attention via *ASM* and *GS*. By **re-weighting the modal hierarchy** within the attention

mechanism, AlignVid improves semantic alignment with negligible computational overhead and minimal aesthetic impact. (iii) **Benchmark-OmitI2V.** We curate a benchmark with 367 human-annotated cases spanning modification, addition, and deletion scenarios.

## 2. Related Works

**Image-to-Video Models.** I2V generation models can be broadly classified into GAN-based, Stable Diffusion-based, and DiT-based paradigms. GAN-based methods (Tulyakov et al., 2017; Skorokhodov et al., 2022) often suffer from inherent challenges in modeling long-term dependencies and high-frequency details. Stable Diffusion-based models leverage UNet architectures. VideoComposer (Wang et al., 2023a) first integrates image conditioning into 3D-UNet by concatenating clean image latents with noisy video latents. Building on this, SVD (Blattmann et al., 2023) and Dynami-Crafter (Xing et al., 2024) inject CLIP (Radford et al., 2021) features from reference images into the denoising process to enhance guidance. Further works explore cascading diffusion framework (Zhang et al., 2023) and leverage first and last frames to improve temporal coherence (Chen et al., 2023; Zeng et al., 2024). DiT-based methods (Brooks et al., 2024; Yang et al., 2024; Polyak et al., 2024; Ma et al., 2024; Kong et al., 2025; Wan et al., 2025) replace U-Net with Transformers by partitioning latent space frame patches into tokens for unified modeling of long-range dependencies. Recent advances (Chen et al., 2025; Zhang & Agrawala, 2025; Xu et al., 2024; Kong et al., 2025; Wan et al., 2025) employ multimodal fusion to align visual and text inputs. **Image-to-Video Generation Benchmarks.** Existing benchmarks for I2V generation primarily focused on evaluating the quality

and consistency of the generated videos. VBench (Huang et al., 2024) introduces comprehensive suites for assessing models. In contrast, AIGCBench (Fan et al., 2023) and EvalCrafter (Liu et al., 2024b) focus on aspects such as text-video alignment and aesthetic quality. Other works have targeted more specific attributes of I2V generation. For instance, temporal compositionality (Feng et al., 2024), visual consistency (Wang et al., 2025) and precise motion control (Ren et al., 2024; Zhang et al., 2025). While existing benchmarks assess overall video quality and alignment, they do not capture *semantic negligence*, i.e., failures to follow instructions for modification or addition.

## 3. Pilot Observation

We investigate the semantic fidelity challenge using the **OmitI2V** benchmark (details in Section 6). We summarize two empirical observations:

**Observation 1: Semantic negligence is prevalent in TI2V.** As summarized in Table 1, state-of-the-art methods often preserve the reference image semantics instead of implementing the requested changes, indicating a misalignment between the textual instruction and the generated video.

**Observation 2: Image perturbations purify the attention landscape by filtering visual redundancy.** Our quantitative study (Figure 2 (c)) reveals that Gaussian blur reshapes the cross-attention distribution. Crucially, we observe a **universal sharpening of attention focus**: after blurring, the **intensity of attention toward core tokens** in the image, video, and text modalities all increase, while their respective entropies consistently decrease. This suggests that by erasing low-level visual details, the model is no longer distracted by redundant visual tokens, leading to a more decisive matching process. Two key nuances further clarify the mechanism of *visual dominance*: (i) **Textual Liberation:** While the focus sharpens across all modalities, the attention weight assigned to *text tokens* shows the most substantial increase. This indicates that the high-fidelity image originally exerted a suppressive effect on textual constraints; once this dominance is weakened, the model becomes significantly more responsive to the prompt. (ii) **Structural Sharpness:** The entropy reduction is most pronounced in the image and video modalities, whereas text entropy remains relatively stable. This implies that text tokens, which are inherently semantically dense, were previously overshadowed by the high-entropy noise of the image.

## 4. Attention Analysis

We adopt a single attention head and studies how scaling the logits of different key groups (text, image, video) affects the attention distribution. At denoising step $t$, we write

$Q_t \in \mathbb{R}^{n \times d}, \quad K_t \in \mathbb{R}^{m \times d}, \quad V_t \in \mathbb{R}^{m \times d_v}$ and define

$$Z_t = \frac{1}{\sqrt{d}} Q_t K_t^\top, \qquad \text{Attn}(Q_t, K_t, V_t) = \sigma(Z_t) V_t, \quad (1)$$

where $\sigma(\cdot)$ denotes the row-wise softmax. Video queries can attend to keys from text, image, and video tokens. We denote a disjoint partition of key indices by

$$\mathcal{I}_{\text{text}}, \mathcal{I}_{\text{img}}, \mathcal{I}_{\text{vid}} \subseteq \{1, \dots, m\}, \qquad (2)$$

and write $K_t = [K_t^{\text{text}}; K_t^{\text{img}}; K_t^{\text{vid}}]$ (up to permutation), which covers both standard DiT and MMDiT architectures. In TI2V, the three groups play different roles: text tokens encode the desired edit, image tokens encode the input frame prior, and video tokens enforce temporal smoothness.

**Energy view and entropy.** For the $i$-th query, let $z^{(i)} \in \mathbb{R}^m$ be the corresponding logits. The log-partition and attention distribution are

$$\Phi(z^{(i)}) = \log \sum_j e^{z_j^{(i)}}, \qquad p^{(i)} = \nabla_{z^{(i)}} \Phi = \sigma(z^{(i)}), \quad (3)$$

with Hessian $\nabla_{z^{(i)}}^2 \Phi = \text{Diag}(p^{(i)}) - p^{(i)} p^{(i)^\top} \succeq 0$ which characterizes the sensitivity of attention probabilities to logit perturbations. To quantify uncertainty within a subset of keys $S \subseteq \{1, \dots, m\}$, we define the restricted softmax and its entropy under inverse temperature $\alpha > 0$ as

$$p_{S,j}^{(i)}(\alpha) = \frac{e^{\alpha z_j^{(i)}}}{\sum_{k \in S} e^{\alpha z_k^{(i)}}}, \ H_{i,S}(\alpha) = -\sum_{j \in S} p_{S,j}^{(i)}(\alpha) \log p_{S,j}^{(i)}(\alpha). \quad (4)$$

### 4.1. Temperature View of Q/K Scaling

**Lemma 4.1** (Q/K scaling as temperature control). *Consider scaling the query or key embeddings by a positive scalar $\gamma_t > 0$. Replacing $Q_t$ by $\gamma_t Q_t$ (or $K_t$ by $\gamma_t K_t$) yields*

$$Z_t' = \frac{1}{\sqrt{d}} Q_t' K_t^\top = \gamma_t Z_t \quad (or\ Z_t' = \frac{1}{\sqrt{d}} Q_t K_t'^\top = \gamma_t Z_t), \quad (5)$$

*so each row of the attention uses a softmax with temperature $\alpha_t = \gamma_t$, i.e. $p^{(i)}(\alpha_t) = \sigma(\alpha_t z^{(i)})$.*

In multi-modal attention, we are interested in scaling only conditioning tokens. Let $S_{\text{cond}} = \mathcal{I}_{\text{text}} \cup \mathcal{I}_{\text{img}}$ denote the conditioning block, and keep video keys unscaled. Conceptually, increasing the temperature on $S_{\text{cond}}$ both increases the total attention mass allocated to conditioning tokens relative to video self-attention and reshapes how attention is distributed within the conditioning block.

### 4.2. Entropy and Semantic Fidelity

**Lemma 4.2** ( Within-block entropy monotonicity). *For any query $i$, subset $S$ of key, and $\alpha > 0$,*

$$\frac{\mathrm{d}}{\mathrm{d}\alpha} H_{i,S}(\alpha) = -\alpha \operatorname{Var}_{p_S^{(i)}(\alpha)}[z_S^{(i)}] \leq 0, \qquad (6)$$

*where the variance is taken with respect to $p_S^{(i)}(\alpha)$. Thus increasing $\alpha$ monotonically reduces the entropy within $S$ unless the logits $\{z_j^{(i)} : j \in S\}$ are degenerate.*

Taking $S = S_{\text{cond}}$ shows that increasing $\alpha_t^{\text{cond}}$ yields a more

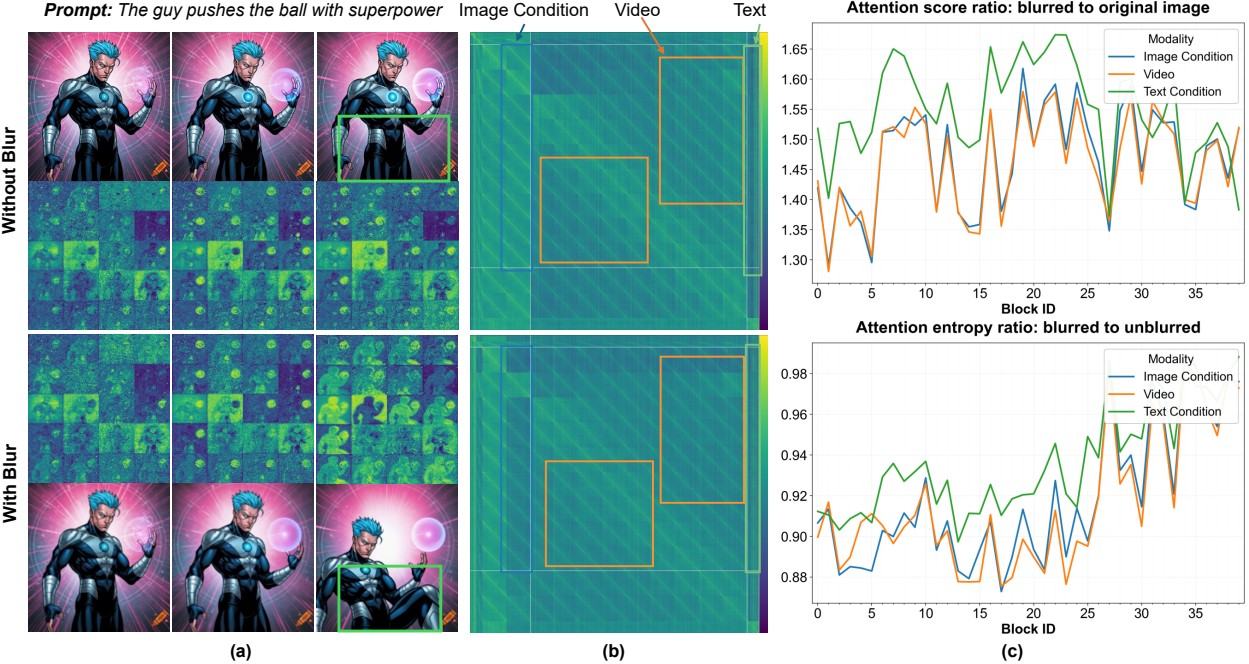

*Figure 2.* **Pilot example.** **(a)** Videos and attention maps generated from the original input image (top) and from the same image after applying a Gaussian blur (bottom). **(b) Attention Visualization:** In the original setting, the model exhibits *visual dominance*, where excessive focus on the reference image suppresses textual constraints and temporal dynamics. Applying blur weakens this dominance, shifting attention toward text tokens and temporal neighbors. **(c) Quantitative Analysis:** Statistics across 30 samples reveal that applying blur leads to a universal increase in attention scores and a universal reduction in entropy across image, video, and text modalities.

concentrated attention distribution over conditioning tokens for each video query, i.e., it reduces the uncertainty about which conditioning tokens the query attends to, while leaving video self-attention unchanged.

**TI2V semantic fidelity.** Mathematically, entropy reduction is the direct consequence of increasing the inverse temperature. From a signal-level viewpoint, the same temperature scaling acts as a *semantic sharpening* operation on the softmax: as $\alpha$ increases, probability mass is reallocated from low-logit tokens to a few high-logit tokens that carry stronger semantic evidence. In our TI2V setting, semantic negligence manifests as a signal imbalance, where attention overemphasizes the image prior and underweights edit-related text and temporal cues, leading the model to preserve the input frame instead of realizing the requested edit. Softmax sharpening (theoretically linked to lower attention entropy) resolves this conflict by scaling relevant token block logits. This forces video queries to shift focus from dominant image cues to amplified text signals, directly boosting semantic adherence.

**Curvature and over-concentration.** For completeness, consider scaling all logits of a query by a common factor $\alpha > 0$ and define $\Phi_i(\alpha) = \Phi(\alpha z^{(i)})$ with Hessian

$$\mathcal{H}_i(\alpha) = \nabla^2_{z^{(i)}} \Phi(\alpha z^{(i)})$$
$$= \alpha^2 \Big( \mathrm{Diag}(p^{(i)}(\alpha)) - p^{(i)}(\alpha) p^{(i)}(\alpha)^\top \Big). \quad (7)$$

where $p^{(i)}(\alpha) = \sigma(\alpha z^{(i)})$. If $\Delta_i$ denotes the gap between the largest and second-largest logits in $z^{(i)}$, one can show that for sufficiently large $\alpha$ the spectral norm $\|\mathcal{H}_i(\alpha)\|_{\mathrm{spec}}$ eventually decreases and converges to zero (proof in the Appendix). Intuitively, very large temperatures collapse attention onto a single token and flatten the energy landscape along off-peak directions.

**Design implications for TI2V.** Based on the above analysis, we summarize the design principles that guide our method. (i) **Temperature as an attention gain knob.** Scaling $Q$ or $K$ is exactly inverse-temperature control and thus offers an explicit way to strengthen or weaken the influence of selected token groups without modifying the inputs. (ii) **Entropy reduction as decisive semantic selection.** Increasing the temperature on a token block reduces its internal entropy and sharpens attention onto a small set of high-logit, semantically relevant tokens.

## 5. Method

Building on the above analysis, we propose **AlignVid**, a training-free approach for modulating attention distributions. AlignVid has two components: (i) **Attention Scaling Modulation (ASM)**, which sharpens prompt-relevant attention; and (ii) **Guidance Scheduling (GS)**, which selectively applies ASM across blocks and denoising steps to preserve

visual fidelity while improving semantic adherence. The method adds negligible overhead. The pseudocode is provided in Algorithm 1 and Algorithm 2 (*Appendix*).

## 5.1. Attention Scaling Modulation

A straightforward way to sharpen attention is to inject external masks. However, this has three drawbacks: (i) masks are static and misaligned with the evolving denoising dynamics; (ii) in open-vocabulary settings, defining reliable masks (e.g., for unseen objects) is brittle; and (iii) maintaining and applying masks adds inference overhead. To overcome these limitations, we introduce **Attention Scaling Modulation (ASM)**, which directly modifies the attention computation by scaling the **query** or **key** embeddings within attention layers. The choice of modulation target depends on the model architecture: in **MMDiT**, where conditioning is injected via query tokens, we modulate $Q$; in **DiT-style cross-attention**, we modulate $K$. Formally, let $Q \in \mathbb{R}^{n_q \times d_k}$, $K \in \mathbb{R}^{n_k \times d_k}$, and $V \in \mathbb{R}^{n_k \times d_v}$. ASM modifies attention by scaling the query or key embeddings before the attention:

$$\text{Attention}_{\text{ASM}}(Q, K, V) = \text{softmax}\left(\frac{Q'(K')^T}{\sqrt{d_k}}\right) V, \quad (8)$$

where $Q'$ and $K'$ are the modulated embeddings. By *Lemma 4.1*, such scaling is equivalent to reparameterizing the row-wise softmax via its inverse temperature $\alpha$.

**(S1) Scalar scaling.** Apply a multiplicative scalar $\gamma_s > 1$ to either $Q$ or $K$:

$$Q' = \gamma_s Q \quad \text{or} \quad K' = \gamma_s K. \quad (9)$$

This sharpens the attention by amplifying the contrast between relevant and irrelevant regions.

**(S2). Energy-based scaling.** Inspired by the energy interpretation of attention, we adaptively set the scaling coefficient according to the sharpness of the logits:

$$\gamma_e = f\left(\frac{1}{n_q n_k} \sum_{i,j} \frac{Q_i K_j^\top}{\sqrt{d_k}}\right), \quad (10)$$

where $f(\cdot)$ is a monotonic function (e.g., sigmoid rescaling) and $n_q, n_k$ denote query/key counts. This encourages stronger modulation when attention logits are diffuse.

## 5.2. Guidance Scheduling

While ASM enhances semantic consistency, applying it indiscriminately across all blocks and steps downgrade perceptual quality. We therefore introduce **Guidance Scheduling (GS)**, which gates ASM at the block and step level.

**Block-level Guidance Scheduling (BGS).** We observe that different transformer blocks contribute unequally: some focus more on foreground semantics, while others capture background context. We selectively apply attention modulation only to *foreground-sensitive* blocks. To identify

*foreground-sensitive* blocks, we perform a lightweight calibration: collect attention maps on a small validation set, project them via PCA to capture dominant directions, and use an off-the-shelf grounding model to separate foreground from background. For each block $l$, we compute its *foreground ratio* $r^{(l)}$, the average fraction of attention mass allocated to foreground tokens. Blocks with $r^{(l)} > \tau$ (0.5) are deemed foreground-sensitive. We assign each block a scaling coefficient:

$$g^{(l)} = \begin{cases} \gamma & \text{if } r^{(l)} > \tau \\ 1 & \text{otherwise,} \end{cases} \quad (11)$$

where $\gamma > 1$ controls the perturbation strength. The modulated attention is then applied:

$$\text{Attention}^{(l)}(Q, K, V) = \text{softmax}\left(\frac{Q(g^{(l)}K)^\top}{\sqrt{d_k}}\right) V. \quad (12)$$

Empirically, we find that most foreground-sensitive blocks lie in the earlier half of the network. Consequently, we consider two variants of BGS in our experiments: (i) using the calibrated set of blocks with $r^{(l)} > \tau$, and (ii) a simpler heuristic that applies modulation to the first 50% of blocks.

**Step-level Guidance Scheduling (SGS).** We further specify *when* modulation is applied along the denoising process. Early steps operate under high noise and determine global semantic alignment, mid steps refine coarse structures, while late steps mainly enhance visual details. Formally, let $t \in \{1, 2, \ldots, T\}$ denote the denoising step. We define a scheduling function:

$$m(t) = \begin{cases} 1 & \text{if } t \in [t_{\text{low}}, t_{\text{high}}], \\ 0 & \text{otherwise,} \end{cases} \quad (13)$$

where $[t_{\text{low}}, t_{\text{high}}]$ denotes the interval of active guidance. To account for implementation differences (scaling either queries or keys), we combine block and step scheduling with an explicit scaling target. Let $s_Q, s_K \in \{0, 1\}$ indicate whether we scale queries or keys ($s_Q + s_K = 1$). We define:

$$g^{(l,t)} = m(t) b^{(l)} (\gamma - 1), \quad (14)$$

where $b^{(l)}$ is the block gate and $m(t) \in \{0, 1\}$ the step mask. Then:

$$Q'^{(l,t)} = (1 + s_Q \times g^{(l,t)}) Q^{(l)}, \ K'^{(l,t)} = (1 + s_K \times g^{(l,t)}) K^{(l)}. \quad (15)$$

The scheduled attention is:

$$\text{Attention}_t^{(l)} = \text{softmax}\left(\frac{Q'^{(l,t)}(K'^{(l,t)})^\top}{\sqrt{d_k}}\right) V^{(l)}. \quad (16)$$

## 6. OmitI2V Benchmark

Existing image-to-video (I2V) benchmarks either lack explicit textual conditioning or assess only coarse text-image consistency, providing limited signal for fine-grained semantic fidelity. We introduce **OmitI2V**, a benchmark designed to evaluate whether TI2V models faithfully execute textual instructions that require *explicit visual edits* to the input

| Method | Semantic Alignment Evaluation | | | Visual Quality Evaluation | |
|---|---|---|---|---|---|
| | Modification ↑ | Addition ↑ | Deletion ↑ | Dynamic Degree ↑ | Aesthetic Quality ↑ |
| Hunyuan I2V (Kong et al., 2025) | 63.28 | 60.34 | 61.94 | 17.74 | 62.04 |
| Wan 2.1 (Wan et al., 2025) | 72.35 | 71.75 | 63.13 | 46.02 | 63.12 |
| Skyreels-v2-I2V (Chen et al., 2025) | 70.02 | 76.64 | 62.95 | 51.16 | 58.94 |
| Skyreels-v2-DF (Chen et al., 2025) | 71.10 | 73.28 | 65.35 | 47.30 | 61.10 |
| FramePack (Zhang & Agrawala, 2025) | 64.99 | 68.55 | 58.14 | 20.05 | 63.94 |
| FramePack F1 (Zhang & Agrawala, 2025) | 64.45 | 67.79 | 58.50 | 24.42 | 63.10 |
| EasyAnimate (Xu et al., 2024) | 65.53 | 67.18 | 60.89 | 45.76 | 61.41 |

*Table 1.* Quantitative comparison on OmitI2V benchmark.

| Method | Semantic Alignment Evaluation | | | ViCLIP Score | | | Visual Quality Evaluation | |
|---|---|---|---|---|---|---|---|---|
| | Modification ↑ | Addition ↑ | Deletion ↑ | Modification ↑ | Addition ↑ | Deletion ↑ | Dynamic Degree ↑ | Aesthetic Quality ↑ |
| FramePack | 64.99 | 68.55 | 58.14 | 20.83 | 21.08 | 20.43 | 20.05 | 63.94 |
| FramePack + Ours | 68.22 (+3.23) | 73.13 (+4.58) | 60.21 (+2.07) | 21.25 (+0.42) | 22.08 (+0.83) | 20.86 (+0.43) | 28.53 (+8.48) | 63.57 (−0.37) |
| FramePack F1 | 64.45 | 67.79 | 58.50 | 21.06 | 19.91 | 20.61 | 24.42 | 63.10 |
| FramePack F1 + Ours | 71.27 (+6.82) | 71.60 (+3.81) | 61.06 (+2.56) | 21.78 (+0.72) | 21.04 (+1.13) | 20.99 (+0.38) | 33.16 (+8.74) | 62.10 (−1.00) |
| Wan2.1 | 72.35 | 71.75 | 63.13 | 20.93 | 20.59 | 20.82 | 46.02 | 63.12 |
| Wan2.1 + Ours | 77.20 (+4.85) | 79.54 (+7.79) | 69.47 (+6.34) | 22.19 (+1.26) | 23.30 (+2.71) | 21.29 (+0.47) | 47.04 (+1.02) | 61.63 (−1.49) |

*Table 2.* **Effectiveness of our method.** Values in parentheses indicate relative improvement (%) over the corresponding baseline.

| Method | Semantic Alignment Evaluation | | | ViCLIP Score | | | Visual Quality Evaluation | |
|---|---|---|---|---|---|---|---|---|
| | Modification ↑ | Addition ↑ | Deletion ↑ | Modification ↑ | Addition ↑ | Deletion ↑ | Dynamic Degree ↑ | Aesthetic Quality ↑ |
| | *FramePack* | | | | | | | |
| Original | 64.99 | 68.55 | 58.14 | 20.83 | 21.08 | 20.43 | 20.05 | 63.94 |
| Scalar scaling | **67.15** | **73.44** | **59.86** | **21.38** | **22.03** | **21.05** | **28.28** | 63.41 |
| Energy-based modulation | 66.61 | 72.37 | 58.66 | 21.26 | 21.79 | 20.76 | 26.48 | **63.62** |
| | *Wan2.1* | | | | | | | |
| Original | 72.35 | 71.75 | 63.13 | 20.93 | 20.59 | 20.82 | 46.02 | 63.12 |
| Scalar scaling | **72.53** | **80.76** | **70.33** | **22.28** | **23.50** | **21.26** | **53.21** | 62.38 |
| Energy-based modulation | 72.40 | 75.65 | 67.86 | 21.56 | 21.82 | 20.97 | 48.90 | **62.67** |

*Table 3.* **Ablation about modulation variants.** Bold values denote the best performance.

image (modification, addition, deletion).

**Evaluation axes.** OmitI2V evaluates two complementary axes. (i) *Semantic Alignment Evaluation* evaluates whether the generated video realizes the prompt-specified edit under the three scenarios. We assess edit-level compliance with a VQA-based yes/no protocol and report accuracy. (ii) *Visual Quality Evaluation* reports the *dynamic degree* (the extent of motion) and *aesthetic quality* (perceptual fidelity and visual appeal), independent of semantic correctness.

**Data and protocol.** The benchmark contains **367** image–text pairs spanning diverse styles (real, synthetic, animation). Each pair is annotated with an edit type (*addition*, *deletion*, or *modification*) that specifies the intended visual change. Conventional metrics such as FVD are not designed to capture edit-level semantic compliance. Instead, for each video, we introduce yes/no questions derived from the prompt and edit type (e.g., *"Did a sunflower appear in front of the house?"*) and compute accuracy using *Qwen2.5-VL-32B* (Wang et al., 2024). We employ the ViCLIP

score (Wang et al., 2023b) as a text-video matching metric.

## 7. Experiments

### 7.1. Comparison Experiments

**Semantic negligence remains prevalent.** Table 1 shows results on OmitI2V-Bench, indicating no existing TI2V model uniformly handles all edit types. For instance, *Wan2.1* achieves the highest accuracy in modification and addition but drops notably in deletion; *Skyreels-v2-I2V* excels at addition yet lacks consistency elsewhere. *FramePack* (and its variant), despite autoregressive priors, exhibits the weakest semantic fidelity, especially in deletion. These results confirm that semantic negligence persists across methods.

**Semantics-aesthetic trade-off.** Table 1 shows that stronger prompt adherence is not necessarily aligned with higher visual quality. For instance, *EasyAnimate* and *Skyreels-v2-DF* attain competitive dynamic degree and aesthetic scores, yet exhibit semantic omissions.

| Method | Semantic Alignment Evaluation | | | ViCLIP Score | | | Visual Quality Evaluation | |
|---|---|---|---|---|---|---|---|---|
| | Modification ↑ | Addition ↑ | Deletion ↑ | Modification ↑ | Addition ↑ | Deletion ↑ | Dynamic Degree ↑ | Aesthetic Quality ↑ |
| *FramePack* | | | | | | | | |
| - | 64.99 | 68.55 | 58.14 | 20.83 | 21.08 | 20.43 | 20.05 | 63.94 |
| Key-image | 64.45 | 61.07 | 52.93 | 14.55 | 11.95 | 15.19 | 6.94 | 24.63 |
| Key-text | 65.71 | 62.90 | **65.81** | 15.88 | 12.81 | 16.10 | 14.91 | 24.81 |
| Key-image and Key-text | **68.22** | **73.13** | 60.21 | **21.25** | **22.08** | **20.86** | **28.53** | **63.57** |
| *Wan2.1* | | | | | | | | |
| - | 72.35 | 71.75 | 63.13 | 20.93 | 20.59 | 20.82 | 46.02 | 63.12 |
| Key in Self-attention | 67.32 | 66.87 | 65.18 | 19.20 | 16.91 | 19.03 | 58.87 | 47.10 |
| Query-image | 76.48 | 80.46 | 65.18 | 22.10 | 23.39 | **21.69** | 51.67 | 61.20 |
| Key-image | 69.48 | 75.42 | 64.49 | 21.14 | 21.41 | 21.03 | 48.33 | 62.55 |
| Query-text | 72.71 | 77.25 | 68.27 | 22.13 | 22.78 | 21.45 | 59.90 | 61.62 |
| Key-text | 71.45 | 78.17 | 67.41 | 21.82 | 22.88 | 21.43 | 55.53 | 61.46 |
| Key-image and Query-text | **76.66** | 79.85 | 67.75 | 21.04 | 21.80 | 21.00 | **60.67** | 61.58 |
| Key-image and Key-text | 73.79 | 78.18 | 66.04 | 22.06 | 23.20 | 21.48 | 43.19 | **62.86** |
| Query-image and Key-text | 72.53 | **80.76** | **70.33** | **22.28** | **23.50** | 21.26 | 53.21 | 62.38 |

*Table 4.* **Ablation on scaling positions.** For *FramePack*, image and text tokens are concatenated and processed via self-attention, making scaling $Q$ or $K$ effectively equivalent (we scale $K$ in practice). For *Wan2.1*, video tokens use self-attention (treated as in *FramePack*), while image and text act as cross-attention conditions where $Q$ and $K$ differ and must be analyzed separately.

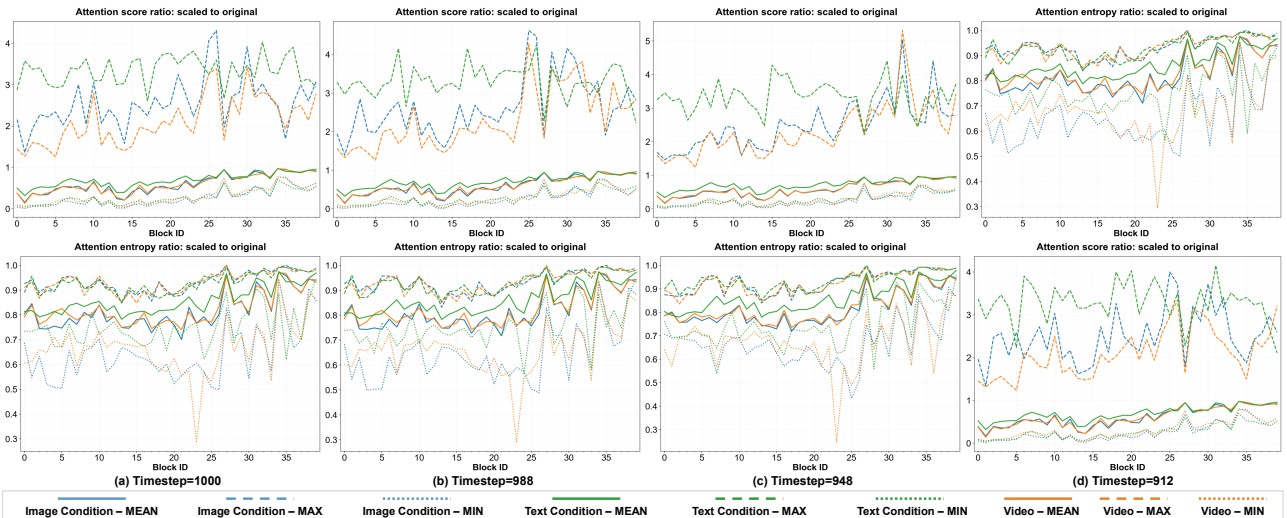

*Figure 3.* **Attention analysis.** ASM sharpens attention (lower entropy), boosts focus on text tokens and suppresses image regions.

**Effectiveness of AlignVid.** Table 2 shows that plugging **AlignVid** into *FramePack*, *FramePack-F1*, and *Wan2.1* yields consistent gains in semantic fidelity and dynamic degree across all edit types, indicating good architectural generality. While aesthetic quality scores may drop slightly, the decrease is minor relative to the substantial improvements in semantic fidelity and motion coherence, validating the design of selective attention scaling and scheduling.

## 7.2. Ablation and Generalization Experiment

**Ablation on modulation strategy.** Table 3 compares the variants: *scalar scaling* and *energy-based modulation*. Both improve semantic fidelity, confirming that attention reweighting is effective. The energy-based variant yields smaller drops in aesthetic quality but also smaller semantic

gains. Considering its additional inference overhead, we adopt *scalar scaling* for other experiments.

**Ablation on scaling position.** We ablate scaling sites inside attention (queries ($Q$) and keys ($K$)) and their image/text partitions (Table 4). On *FramePack*, where image and text tokens are concatenated and processed via self-attention, scaling $Q$ or $K$ is effectively equivalent; empirically, combining image- and text-side key scaling delivers the strongest overall semantic gains. In contrast, key-image only provides limited benefits and can hurt aesthetic metrics. On *Wan2.1*, where video tokens use self-attention but image/text act as cross-attention conditions, positions are no longer symmetric: pairing image queries with text keys attains the best *addition/deletion* accuracy, pairing image keys with text queries yields the highest dynamic degree,

| BGS | SGS | Semantic Alignment Evaluation | | | ViCLIP Score | | | Visual Quality Evaluation | |
|---|---|---|---|---|---|---|---|---|---|
| | | Modification ↑ | Addition ↑ | Deletion ↑ | Modification ↑ | Addition ↑ | Deletion ↑ | Dynamic Degree ↑ | Aesthetic Quality ↑ |
| *FramePack* | | | | | | | | | |
| - | - | 64.99 | 68.55 | 58.14 | 20.83 | 21.08 | 20.43 | 20.05 | 63.94 |
| All | Early Steps | 67.15 | 73.44 | 59.86 | 21.38 | 22.03 | 21.05 | 28.28 | 63.41 |
| All | Middle Steps | 62.71 | 70.01 | 56.60 | 20.85 | 21.06 | 20.54 | 20.05 | 63.96 |
| All | End Steps | 64.63 | 69.62 | 57.63 | 20.80 | 21.08 | 20.47 | 19.54 | 63.94 |
| All | All | 69.84 | 76.03 | 59.86 | 21.56 | 22.30 | 21.31 | 32.13 | 61.56 |
| Foreground-focus | Early Steps | 68.22 | 73.13 | 60.21 | 21.25 | 22.08 | 20.86 | 28.53 | 63.57 |
| Background-focus | Early Steps | 66.25 | 69.16 | 56.26 | 20.88 | 21.03 | 20.50 | 17.99 | 64.02 |
| First half blocks | Early Steps | 66.61 | 73.89 | 58.31 | 20.68 | 22.16 | 20.76 | 25.96 | 63.58 |
| Last half blocks | Early Steps | 65.35 | 69.92 | 57.18 | 20.39 | 21.19 | 20.56 | 22.11 | 63.49 |
| *Wan2.1* | | | | | | | | | |
| - | - | 72.35 | 71.75 | 63.13 | 20.3 | 21.08 | 20.43 | 46.02 | 63.12 |
| All | Early Steps | 72.53 | 80.76 | 70.33 | 22.28 | 23.50 | 21.26 | 53.21 | 61.38 |
| All | Middle Steps | 68.76 | 74.81 | 61.41 | 21.37 | 21.22 | 20.89 | 42.16 | 62.91 |
| All | End Steps | 69.84 | 74.05 | 66.90 | 21.20 | 21.86 | 21.44 | 53.98 | 61.55 |
| All | All | 78.28 | 80.46 | 69.13 | 22.63 | 24.26 | 21.93 | 49.36 | 60.59 |
| Foreground-focus | Early Steps | 77.20 | 79.54 | 69.47 | 22.19 | 23.30 | 21.29 | 47.04 | 61.63 |
| Background-focus | Early Steps | 71.99 | 75.88 | 65.35 | 21.26 | 21.93 | 20.86 | 41.90 | 62.62 |
| First half blocks | Early Steps | 76.55 | 78.85 | 68.10 | 22.18 | 22.60 | 21.40 | 52.70 | 61.54 |
| Last half blocks | Early Steps | 73.68 | 77.89 | 62.64 | 20.63 | 22.48 | 21.20 | 50.31 | 61.47 |

*Table 5.* **Ablation of block- and step-level guidance scheduling.** Gating ASM to BGS boosts semantic fidelity.

| Method | Semantic Alignment Evaluation | | | ViCLIP Score | | | Visual Quality Evaluation | |
|---|---|---|---|---|---|---|---|---|
| | Modification ↑ | Addition ↑ | Deletion ↑ | Modification ↑ | Addition ↑ | Deletion ↑ | Dynamic Degree ↑ | Aesthetic Quality ↑ |
| CFG=1 (no cfg) | 63.55 | 63.66 | 61.06 | 19.20 | 17.09 | 19.67 | 41.65 | 61.19 |
| CFG=1 + AlignVid | 65.88 | 72. 52 | 60.21 | 19.51 | 18.47 | 19.89 | 42.48 | 62.11 |
| CFG=5 (Official) | 72.35 | 71.75 | 63.13 | 20.83 | 21.08 | 20.43 | 46.02 | 63.12 |
| CFG=5 + AlignVid | 77.20 | 79.54 | 69.47 | 22.19 | 23.30 | 21.29 | 47.04 | 61.63 |

*Table 6.* Comparison with CFG on Wan2.1.

and image keys with text keys offers the best aesthetic score. Overall, jointly modulating image- and text-side sites yields the best semantic–visual trade-off, with architecture-aware preferences between self- and cross-attention.

**Ablation on block- and step-level guidance scheduling.** We evaluate the proposed BGS and SGS strategy on *FramePack* and *Wan2.1*, as shown in Table 5. *For BGS,* limiting ASM to foreground-focused blocks improves semantic fidelity while mitigating aesthetic degradation by concentrating modulation where text–visual grounding is strongest. *For SGS,* activating guidance in early denoising steps yields the largest semantic gains; mid/late activation offers weaker semantic improvements but better preserves aesthetics. Enabling guidance at all steps maximizes semantic fidelity but incurs a noticeable visual quality drop (e.g., a $2.38\%$ relative decrease for *FramePack*). Balancing these trade-offs, we adopt an early-step schedule by default.

**Choice of the scaling factor.** The strength $\gamma$ admits a principled, transferable choice. Lemma 2 (Appendix B.2) shows that scaling monotonically reduces the conditioning entropy, so the semantic–quality trade-off is predictable by design: increasing $\gamma$ improves semantic fidelity while only gradu-

ally lowering aesthetic quality (Appendix H.2). This makes $\gamma=1.35$ transferable across backbones without per-model search, analogous to selecting CFG strength. We further validate a fully automated, per-token *Conflict-Aware Scaling* (CAS) variant that sets $\gamma$ from the image–text attention conflict; it matches fixed scaling on semantics while better preserving aesthetics (Appendix H.3).

**Attention analysis.** We further analyze attention maps before and after applying AlignVid on the benchmark, as illustrated in Figure 3. Concretely, we compute (i) attention distributions over different token groups, (ii) the ratio between the maximum attention scores, and (iii) the ratio of attention entropies for video queries. After modulation, the attention distributions become noticeably sharper, reflected by a consistent decrease in attention entropy. At the signal level, video queries allocate stronger attention to text tokens and temporally adjacent frames, and relatively less to static image regions, encouraging the model to focus more on prompt and temporal cues. This shift in attention patterns correlates well with the improved semantic consistency observed in the generated videos.

**Comparison with Classifier-Free Guidance (CFG).** We

| Method | Single object | Two object | Counting | Colors | Position | Color attribution | Aesthetic Score |
|---|---|---|---|---|---|---|---|
| OmniGen2 (Wu et al., 2025) | 0.99 | 0.94 | 0.67 | 0.85 | 0.55 | 0.62 | 5.517 |
| + AlignVid | 1.00(+0.01) | 0.97(+0.03) | 0.52(-0.15) | 0.89(+0.04) | 0.60(+0.05) | 0.70(+0.08) | 5.568(+0.05) |

*Table 7.* Quantitative results on GenEval. Prompt rewriter is not utilized during inference.

| Model | Subject Consistency | Temporal Style | Temporal Flickering | Spatial Relationship | Scene | Overall Consistency | Object Class | Multiple Objects |
|---|---|---|---|---|---|---|---|---|
| Wan2.1-T2V-1.3B | 94.24 | 22.67 | 99.32 | 72.74 | 19.62 | 23.59 | 79.03 | 53.35 |
| + AlignVid | 94.51(+0.27) | 23.46(+0.79) | 98.66(-0.66) | 84.25(+11.51) | 25.80(+6.18) | 24.47(+0.88) | 79.91(+0.88) | 66.46(+13.11) |

| Model | Motion Smoothness | Imaging Quality | Dynamic Degree | Color | Background Consistency | Appearance Style | Aesthetic Quality |
|---|---|---|---|---|---|---|---|
| Wan2.1-T2V-1.3B | 97.77 | 69.70 | 70.83 | 88.08 | 98.09 | 19.58 | 64.60 |
| + AlignVid | 98.05(+0.28) | 68.53(-1.17) | 68.06(-2.77) | 91.80(+3.72) | 98.20(+0.11) | 20.16(+0.58) | 62.69(-1.91) |

*Table 8.* Quantitative results on VBench. AlignVid also yields gains in the T2V task.

| Model | Add | Adjust | Extract | Replace | Remove | Background | Style | Compose | Action | Aesthetic Score |
|---|---|---|---|---|---|---|---|---|---|---|
| OmniGen2 (Wu et al., 2025) | 2.52 | 3.27 | 2.08 | 3.12 | 2.83 | 3.65 | 4.57 | 2.89 | 4.59 | 5.606 |
| + AlignVid | 3.53(+1.01) | 3.12(-0.15) | 2.04(-0.04) | 3.18(+0.06) | 3.33(+0.50) | 3.65 | 4.75(+0.18) | 2.43(-0.46) | 4.50(-0.09) | 5.624(+0.02) |

*Table 9.* Quantitative results on ImgEdit. AlignVid also yields gains in the image editing task.

also compare the proposed AlignVid with classifier-free guidance (CFG) in Wan2.1. As shown in Table 6, applying AlignVid on top of CFG further improves performance. Compared with CFG, AlignVid enjoys two practical advantages: (i) it requires no additional training, and (ii) it introduces negligible extra inference overhead (Appendix).

**Generalization: AlignVid on text-to-image generation.** We further evaluate the generalization of AlignVid on text-to-image (T2I) generation, using OmniGen2 (Wu et al., 2025) as the baseline. As reported on the GenEval (Ghosh et al., 2024) in Table 7, incorporating AlignVid improves all metrics except *Counting*, indicating that our attention modulation can also transfer to the image domain.

**Generalization: AlignVid on text-to-video (T2V) generation.** We also evaluate AlignVid on T2V generation, using Wan2.1-T2V-1.3B (Wu et al., 2025) with a scale coefficient of 1.35. As reported on the VBench (Huang et al., 2024) benchmark in Table 8, integrating AlignVid improves most dimensions, while leading to decreases in *Temporal Flickering*, *Imaging Quality*, *Dynamic Degree*, and *Aesthetic Quality*. Some metrics appear to be closely coupled: when AlignVid encourages stronger motion and temporal changes, the resulting videos may exhibit mild motion blur, which can hurt perceived sharpness and aesthetic scores, even though the prompt adherence is improved.

**Generalization: AlignVid on image editing.** We also apply it to an image editing benchmark (ImgEdit (Ye et al., 2025)), using OmniGen2 (Wu et al., 2025) as the baseline model. As shown in Table 9, integrating AlignVid results in gains across editing categories, including *Add*, *Replace*, *Remove*, and *Style*, and also improves the aesthetic score. Interestingly, this contrasts with our observations in video

generation, where AlignVid slightly reduces aesthetic quality. A explanation is that, in the video setting, temporal modeling tends to introduce additional motion blur, whereas image editing is not subject to such temporal artifacts.

## 8. Conclusion

In this paper, to mitigate the challenge of *semantic negligence*, we proposed **AlignVid**, a training-free method based on an energy-based perspective of attention. Our analysis links query/key scaling to a flatter energy landscape and a more concentrated attention distribution. The proposed method comprises *ASM* for attention rescaling and *GS* for selective deployment across transformer blocks and denoising steps. To facilitate evaluation, we provide **OmitI2V**, a benchmark consisting of 367 human-annotated samples across three scenarios, namely modification, addition, and deletion. Experiment results show that AlignVid yields consistent improvements in semantic fidelity.

## Impact Statement

Our work improves text-guided image-to-video generation by enhancing prompt adherence via a training-free attention modulation method (AlignVid) and by providing a diagnostic benchmark (OmitI2V) for measuring fine-grained edit compliance; while this can benefit creative and assistive applications by making generation more controllable and reliable, improved controllability may also increase the risk of misuse for producing persuasive misleading or deceptive synthetic videos, so we recommend responsible deployment practices such as disclosure of AI-generated content and adherence to relevant policies and regulations.

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

# Appendix Contents

## A. The Use of Large Language Models

Large language models (LLMs) (Hurst et al., 2024) are used as general-purpose assistants for language polishing (grammar, tone), LaTeX phrasing, and minor reorganization of exposition. LLMs are *not* used to design experiments, generate or label data, or produce claims. The authors take full responsibility for all content.

## B. Detailed Proofs for Attention Scaling Analysis

### B.1. Lemma 1: Q/K Scaling as Temperature Control

**Statement.** Let $Q'_t = \gamma_t Q_t$ and $K'_t = \eta_t K_t$. Then

$$Z'_t = \frac{1}{\sqrt{d}} Q'_t K'^{\top}_t = (\gamma_t \eta_t) Z_t := \alpha_t Z_t, \quad (17)$$

so for the $i$-th row the attention is $p^{(i)}(\alpha_t) = \sigma(\alpha_t z_t^{(i)})$, i.e., a row-wise softmax with inverse temperature $\alpha_t$. In particular, scaling only $Q$ (resp. $K$) yields $\alpha_t = \gamma_t$ (resp. $\alpha_t = \eta_t$).

**Proof.** By definition,

$$Z_t = \frac{1}{\sqrt{d}} Q_t K_t^{\top}, \quad (18)$$

and after scaling,

$$Z'_t = \frac{1}{\sqrt{d}} (\gamma_t Q_t)(\eta_t K_t)^{\top} = (\gamma_t \eta_t) Z_t. \quad (19)$$

For the $i$-th row,

$$\sigma(\alpha_t z_t^{(i)})_j = \frac{\exp(\alpha_t z_{t,j}^{(i)})}{\sum_{k=1}^{m} \exp(\alpha_t z_{t,k}^{(i)})}, \quad (20)$$

which is softmax with inverse temperature $\alpha_t$ (temperature $T = 1/\alpha_t$). $\square$

### B.2. Lemma 2: Entropy Monotonicity Under Scaling

**Statement.** For any query $i$ and $\alpha > 0$,

$$\frac{\mathrm{d}}{\mathrm{d}\alpha} H_i(\alpha) = -\alpha \operatorname{Var}_{p^{(i)}(\alpha)}[z^{(i)}] \leq 0. \quad (21)$$

**Proof.** Let

$$p_j^{(i)}(\alpha) = \frac{e^{\alpha z_j^{(i)}}}{\sum_k e^{\alpha z_k^{(i)}}}, \quad (22)$$

and write the entropy as

$$H_i(\alpha) = \log \sum_j e^{\alpha z_j^{(i)}} - \alpha \sum_j p_j^{(i)}(\alpha) z_j^{(i)}. \quad (23)$$

Define $\mu(\alpha) = \sum_j p_j^{(i)}(\alpha) z_j^{(i)} = \mathbb{E}_{p^{(i)}(\alpha)}[z^{(i)}]$. Then

$$\frac{\mathrm{d}}{\mathrm{d}\alpha} \log \sum_j e^{\alpha z_j^{(i)}} = \frac{\sum_j z_j^{(i)} e^{\alpha z_j^{(i)}}}{\sum_k e^{\alpha z_k^{(i)}}} = \mu(\alpha). \quad (24)$$

Using $\frac{\partial p_j}{\partial \alpha} = p_j(z_j^{(i)} - \mu(\alpha))$,

$$\begin{aligned} \frac{\mathrm{d}}{\mathrm{d}\alpha} \mu(\alpha) &= \sum_j z_j^{(i)} \frac{\partial p_j}{\partial \alpha} \\ &= \sum_j z_j^{(i)} p_j (z_j^{(i)} - \mu(\alpha)) \\ &= \mathbb{E}_p[z^{(i)2}] - \mu(\alpha)^2 \\ &= \operatorname{Var}_p[z^{(i)}]. \end{aligned} \quad (25)$$

Therefore,

$$\begin{aligned} \frac{\mathrm{d}}{\mathrm{d}\alpha} H_i(\alpha) &= \mu(\alpha) - \left( \mu(\alpha) + \alpha \operatorname{Var}_p[z^{(i)}] \right) \\ &= -\alpha \operatorname{Var}_p[z^{(i)}] \leq 0. \end{aligned} \quad (26)$$

Equality holds iff the row logits are degenerate (zero variance). $\square$

### B.3. Theorem: Asymptotic Curvature Decay Under Scaling

**Statement.** Let $p^{(i)}(\alpha) = \sigma(\alpha z^{(i)})$ and

$$\begin{aligned} H_i(\alpha) &= \nabla_{z^{(i)}}^2 \Phi(\alpha z^{(i)}) \\ &= \alpha^2 \left( \operatorname{Diag}(p^{(i)}(\alpha)) - p^{(i)}(\alpha) p^{(i)}(\alpha)^{\top} \right). \end{aligned} \quad (27)$$

Let $j^{\star} = \arg\max_j z_j^{(i)}$ and $\Delta_i = z_{j^{\star}}^{(i)} - \max_{j \neq j^{\star}} z_j^{(i)} > 0$. Then there exists $\alpha_{\star} = \alpha_{\star}(\Delta_i, m)$ such that for all $\alpha \geq \alpha_{\star}$,

$$\frac{\mathrm{d}}{\mathrm{d}\alpha} \|H_i(\alpha)\|_{\mathrm{spec}} \leq 0, \qquad \lim_{\alpha \to \infty} \|H_i(\alpha)\|_{\mathrm{spec}} = 0. \quad (28)$$

**Proof.** First, a standard softmax gap bound gives

$$\begin{aligned} p_{j^{\star}} &= \frac{1}{1 + \sum_{j \neq j^{\star}} \exp\left(\alpha(z_j^{(i)} - z_{j^{\star}}^{(i)})\right)} \\ &\geq \frac{1}{1 + (m-1)e^{-\alpha \Delta_i}}. \end{aligned} \quad (29)$$

hence, for the tail mass $\varepsilon(\alpha) := 1 - p_{j^{\star}}$,

$$\varepsilon(\alpha) \leq \frac{(m-1)e^{-\alpha \Delta_i}}{1 + (m-1)e^{-\alpha \Delta_i}} \leq (m-1)e^{-\alpha \Delta_i}. \quad (30)$$

Let $C(p) = \operatorname{Diag}(p) - pp^{\top}$ so that $H_i(\alpha) = \alpha^2 C(p)$. For any $i$, $C_{ii} = p_i(1 - p_i)$ and $C_{ij} = -p_i p_j$ for $i \neq j$. By the Gershgorin disk theorem, every eigenvalue $\lambda$ satisfies

$$\lambda \leq \max_i \{C_{ii} + \sum_{j \neq i} |C_{ij}|\} = \max_i \{2 p_i(1 - p_i)\}. \quad (31)$$

When $\alpha$ is large, $p_{j^\star} = 1 - \varepsilon(\alpha)$ and $\sum_{j \neq j^\star} p_j = \varepsilon(\alpha)$, so

$$\max_i p_i(1-p_i) = \max\{(1-\varepsilon)\varepsilon, \max_{j \neq j^\star} p_j(1-p_j)\} \leq \varepsilon(\alpha). \tag{32}$$

Therefore,

$$\begin{aligned} \|C(p)\|_{\text{spec}} &\leq 2\,\varepsilon(\alpha), \\ \|H_i(\alpha)\|_{\text{spec}} &\leq 2\alpha^2(m-1)\,e^{-\alpha\Delta_i}. \end{aligned} \tag{33}$$

The right-hand side tends to 0 as $\alpha \to \infty$, proving the limit. Moreover,

$$\frac{\mathrm{d}}{\mathrm{d}\alpha}\left(\alpha^2 e^{-\alpha\Delta_i}\right) = \alpha e^{-\alpha\Delta_i}(2 - \alpha\Delta_i), \tag{34}$$

which is nonpositive for $\alpha \geq 2/\Delta_i$. Hence there exists $\alpha_\star = \alpha_\star(\Delta_i, m)$ (e.g., $\alpha_\star \geq 2/\Delta_i$) such that $\|H_i(\alpha)\|_{\text{spec}}$ is eventually nonincreasing. $\qquad\square$

**Intuition.** As $\alpha$ grows, softmax mass collapses onto the top logit. The tail mass decays exponentially in $\alpha\Delta_i$, forcing the non-principal directions of to vanish. Although the prefactor $\alpha^2$ can initially increase curvature, the exponential tail dominates asymptotically, so the spectral norm ultimately decreases and converges to zero.

# C. Theoretical Guarantees of Attention Scaling

## C.1. Lipschitz Continuity of Attention Output

We consider the attention output for query $i$ under Q/K scaling factor $\alpha$:

$$y^{(i)}(\alpha) = \sum_{j=1}^{m} p_j^{(i)}(\alpha)V_j = V^\top p^{(i)}(\alpha), \tag{35}$$

where $p^{(i)}(\alpha) = \text{softmax}(\alpha z^{(i)})$.

**Theorem C.1** (Lipschitz Continuity of Attention Output). *For any $\alpha_1, \alpha_2 > 0$, the following bound holds:*

$$\|y^{(i)}(\alpha_1) - y^{(i)}(\alpha_2)\|_2 \leq \frac{1}{2}\|V\|_2\,\|z^{(i)}\|_2\,|\alpha_1 - \alpha_2|. \tag{36}$$

*Detailed Proof.* The derivative of $y^{(i)}(\alpha)$ w.r.t. $\alpha$ is

$$\frac{d}{d\alpha}y^{(i)}(\alpha) = V^\top \frac{d}{d\alpha}p^{(i)}(\alpha). \tag{37}$$

The softmax Jacobian is

$$\begin{aligned} \frac{\partial p_j^{(i)}}{\partial \alpha} &= p_j^{(i)}\left(z_j^{(i)} - \sum_k p_k^{(i)}z_k^{(i)}\right) \\ &= p_j^{(i)}\left(z_j^{(i)} - \mathbb{E}_{p^{(i)}}[z^{(i)}]\right). \end{aligned} \tag{38}$$

Hence, in vector form:

$$\frac{d}{d\alpha}p^{(i)}(\alpha) = \text{Diag}(p^{(i)})z^{(i)} - (p^{(i)}z^{(i)\top})p^{(i)}. \tag{39}$$

It is known that the spectral norm of this softmax derivative is bounded by

$$\left\|\frac{d}{d\alpha}p^{(i)}(\alpha)\right\|_2 \leq \frac{1}{2}\|z^{(i)}\|_2. \tag{40}$$

Finally,

$$\begin{aligned} \left\|\frac{d}{d\alpha}y^{(i)}(\alpha)\right\|_2 &\leq \|V\|_2 \left\|\frac{d}{d\alpha}p^{(i)}(\alpha)\right\|_2 \\ &\leq \frac{1}{2}\|V\|_2\,\|z^{(i)}\|_2. \end{aligned} \tag{41}$$

By the mean value theorem,

$$\|y^{(i)}(\alpha_1) - y^{(i)}(\alpha_2)\|_2 \leq \frac{1}{2}\|V\|_2\,\|z^{(i)}\|_2\,|\alpha_1 - \alpha_2|, \tag{42}$$

proving Lipschitz continuity. $\qquad\square$

**Remark.** This theorem guarantees that scaling Q/K with $\alpha$ produces a bounded change in attention outputs, proportional to the magnitude of $\alpha$ deviation.

## C.2. Impact on a Single Diffusion Step

Consider a single DDIM/ODE update:

$$x_{t-1} = a_t x_t + b_t\,\varepsilon_\theta(x_t, t), \tag{43}$$

where $\varepsilon_\theta$ is $L_y$-Lipschitz in the attention output $y$.

**Proposition C.2** (Upper Bound on State Deviation). *If selective Q/K scaling is applied at step $t$ with factor $\alpha_t$, then the updated state deviation satisfies*

$$\|x'_{t-1} - x_{t-1}\|_2 \leq |b_t|\,L_y \cdot \frac{1}{2}\|V_t\|_2\,\|z_t^{(i)}\|_2\,|\alpha_t - 1|. \tag{44}$$

*Detailed Proof.* Let $\varepsilon'_\theta$ denote the modified noise prediction after scaling. By Lipschitz continuity:

$$\begin{aligned} \|\varepsilon'_\theta - \varepsilon_\theta\|_2 &\leq L_y\|y' - y\|_2 \\ &\leq L_y \cdot \frac{1}{2}\|V_t\|_2\,\|z_t^{(i)}\|_2\,|\alpha_t - 1|. \end{aligned} \tag{45}$$

The diffusion step multiplies this perturbation by $b_t$:

$$\begin{aligned} \|x'_{t-1} - x_{t-1}\|_2 &= |b_t|\,\|\varepsilon'_\theta - \varepsilon_\theta\|_2 \\ &\leq |b_t|\,L_y \cdot \frac{1}{2}\|V_t\|_2\,\|z_t^{(i)}\|_2\,|\alpha_t - 1|. \end{aligned} \tag{46}$$

proving the proposition. $\qquad\square$

**Remark.** This bound ensures that selective attention scaling introduces controlled perturbations, allowing smooth adjustment of semantic fidelity without destabilizing the generation process.

**Design implications for TI2V.** Based on the above analysis, we summarize the design principles that guide our method. (i) **Temperature as an attention gain knob.** Scaling $Q$ or $K$ is exactly inverse-temperature control and thus offers an explicit way to strengthen or weaken the influence of selected token groups without modifying the inputs. (ii) **Entropy reduction as decisive semantic selection.** Increasing the temperature on a token block reduces its internal entropy and sharpens attention onto a small set of high-logit, semantically relevant tokens.

## D. Details of OmitI2V Benchmark

OmitI2V is a benchmark designed to assess the capability of generating videos from images driven by textual instructions, specifically within complex scenarios. Unlike traditional image-to-video tasks, our focus is more on "editing" than "generation". Given an image and a natural language instruction, the model outputs a video that accurately performs the specified *additions, deletions, or modifications*, while preserving the identity, structure, and physical consistency.

**Task Definition.** 1) **Operation Types:** Covering Addition, Deletion, and Modification, representing the most common human interventions in visual media. 2) **Granularity Requirements:** We specify extensible subtypes, ensuring tasks are both diagnostic and diverse. This fine granularity allows for a comprehensive assessment across multiple dimensions.

**Data Construction.** The dataset combines both real and synthetic data: 1) **reference images:** Selected open image or video dataset to ensure high resolution and clear copyright. 2) **Synthetic Enhancement:** Using GPT-4o to generate rare and extreme scenarios (e.g., severe weather, sci-fi effects) to broaden distribution coverage. 3) **Manual Curation and Annotation:** Image-instruction pairs are designed and curated by humans to ensure clear intent.

**Evaluation Methodology.** For evaluating, we employ existing metrics, such as dynamic degree and aesthetic quality in Vbench (Huang et al., 2024), to assess the quality of generated videos. Notably, we do not calculate subject consistency and background consistency, given the nature of adding or removing subjects. Additionally, we introduce Visual Question Answering (VQA), where a Multimodal Large Language Model (MLLM) answers questions based on video content, thereby enhancing the comprehensiveness of the evaluation.

**Attention Analysis in Generative Models.** Attention-based modulation has attracted increasing interest as a method to enable zero-shot image and video editing (Liu et al., 2024a). For image editing, prior work manipulates distinct components of the attention mechanism (Hertz et al., 2022; Cao et al., 2023) to regulate text-image correspondence while preserving geometric and structural properties of the source content (Liu et al., 2024a; Chen et al., 2024). In the video setting, these ideas are extended to enforce temporal consistency across frames: recent methods adapt cross-attention for sequence-level control (Qi et al., 2023; Cai et al., 2025; Jin et al., 2025; Yang et al., 2025) or integrate self-attention with masks derived from cross-attention features (Ma et al., 2025) to steer the generative process. In this work, we examine TI2V prompt adherence from an energy-based perspective and empirically establish a connection between attention distribution and semantic fidelity: lower attention entropy is associated with stronger semantic alignment.

### D.1. Statistical Analysis

Figure 4 summarizes the composition of the OmitI2V benchmark across three axes: edit type, visual domain, and image source. These statistics indicate that the benchmark is well-balanced along the primary task dimension and encompasses a broad spectrum of real-world and synthetic content.

**Edit-type balance.** We enforce near-uniform sampling across the three core types. *Modification* tasks constitute 34.19%, *Addition* tasks 33.16%, and *Deletion* tasks 32.65%. This equilibrium prevents any single operation from dominating the evaluation signal and enables fair comparisons.

**Domain diversity.** We annotate every sample with a fine-grained domain label drawn from the eight mutually-exclusive classes defined below. These labels capture both semantic content and context, enabling granular diagnostics of model robustness. **Living Beings** Any depiction of biological organisms, including but not limited to humans (portraits, crowd scenes, daily activities), domestic and wild animals, and anthropomorphic creatures. The defining criterion is the presence of animate life as the primary subject. **Arts & Entertainment** Creative or performative artifacts that are either hand-drawn or computer-generated, such as cartoons, anime, video-game assets, CGI sequences, virtual idols, and stylized artistic renditions. Realistic photographs of artworks in situ are excluded. **Nature & Environment** Representations of the natural world, spanning landscapes, seascapes, forests, deserts, weather phenomena, macro flora, and non-anthropocentric fauna in their ecological context. Urban parks are classified here only when the natural element dominates the composition. **Structures** Man-made architectural entities, from iconic landmarks and historical edifices to vernacular housing and industrial facilities. Interior shots are included when architectural design is the

focal element. **Objects)** Inanimate physical items, ranging from everyday household articles and consumer products to vehicles, tools, and brand logos. Items are labeled OBJ when they constitute the primary subject rather than mere scene fillers. **Technological & Virtual Elements** Artifacts of modern technology and digital culture, including user-interface screenshots, HUD overlays, AR/VR visualizations, holographic projections, and abstract algorithmic renderings. **Food & Necessities** Edible goods, beverages, cooking processes, and essential daily commodities. Prepared dishes, raw ingredients, and packaged products are all subsumed under this class. **Text & Communication** Static or dynamic textual content designed for human communication, such as signage and logos, provided that text is the dominant visual element.

**Provenance breakdown.** Real photographs dominate the collection (75.58%). Animation frames contribute 18.25%, and purely synthetic images rendered or hallucinated by GPT-4o make up 4.63%. This mix exposes models to both natural statistics and out-of-distribution, synthetic edge cases.

### D.2. Qualitative Visualization of Samples

In this section, we delve into the qualitative visualization of the samples (Figure 5-Figure 10). The description of each sample contains clear expected changes and key elements. This information not only aids in understanding the content depicted in the images but also highlights the critical points of change within the visualization. This interpretive approach ensures the uniformity and accuracy of the sample presentation, allowing each representative change to clearly convey its core concept.

Additionally, the samples are categorized into different main and sub-categories. This organizational method enables a systematic approach to browsing and analyzing the samples. For specific domains, such as human, nature, or animation, this categorization helps us pinpoint and comprehend factors that affect particular types of images.

The questions and answers in the samples further explore various aspects of the images, ranging from action correctness to object presence and dynamic changes. These questions assist in evaluating the standards of the images and their transformations, allowing observers to analyze the sample performance from an evaluative perspective.

### E. Experimental Setup

We evaluate semantic negligence in TI2V generation using our **OmitI2V** benchmark. More experiments, including evaluations on other I2V benchmarks, hyperparameter ablations, efficiency comparisons, and qualitative visualizations, are provided in *Appendix*.

**Baseline models.** We select two representative TI2V models to cover the main architectural lineages: **FramePack** (MM-DiT) (Zhang & Agrawala, 2025) concatenates multi-modal tokens, while **Wan2.1** (DiT) (Wan et al., 2025) factorizes image and text cross-attention.

**Evaluation metrics.** We evaluate semantic negligence in TI2V generation using our **OmitI2V** benchmark. We adopt metrics from VBench (Huang et al., 2024), including dynamic degree and aesthetic quality. To assess semantic alignment, we introduce a Visual Question Answering (VQA) protocol: a multimodal large language model (Qwen2.5-VL-32B) answers questions about the video content, providing an additional, interpretable measure of semantic correctness. We additionally employ the ViCLIP score as a text semantic matching metric for ablation experiments.

### F. Details about Baseline

We select **FramePack** and **Wan2.1** as baselines to cover the two dominant architectural lineages in current diffusion-based video models.

**MM-DiT family.** FramePack instantiates the MM-DiT architecture, which interleaves multi-modal (text–image–video) tokens within a single transformer. Beyond state-of-the-art short-form editing quality, FramePack uniquely supports autoregressive long-video generation; this capability is essential for stress-testing temporal coherence when edits propagate over extended horizons.

**DiT family.** Wan2.1 adopts the standard DiT backbone that factorizes spatial and temporal attention. Its simplicity, parameter efficiency, and widespread adoption make it a representative baseline for the DiT lineage. Together, these two models span the principal design choices—joint versus factorized attention, short versus autoregressive generation—thereby establishing a rigorous and reproducible reference for OmitI2V evaluations.

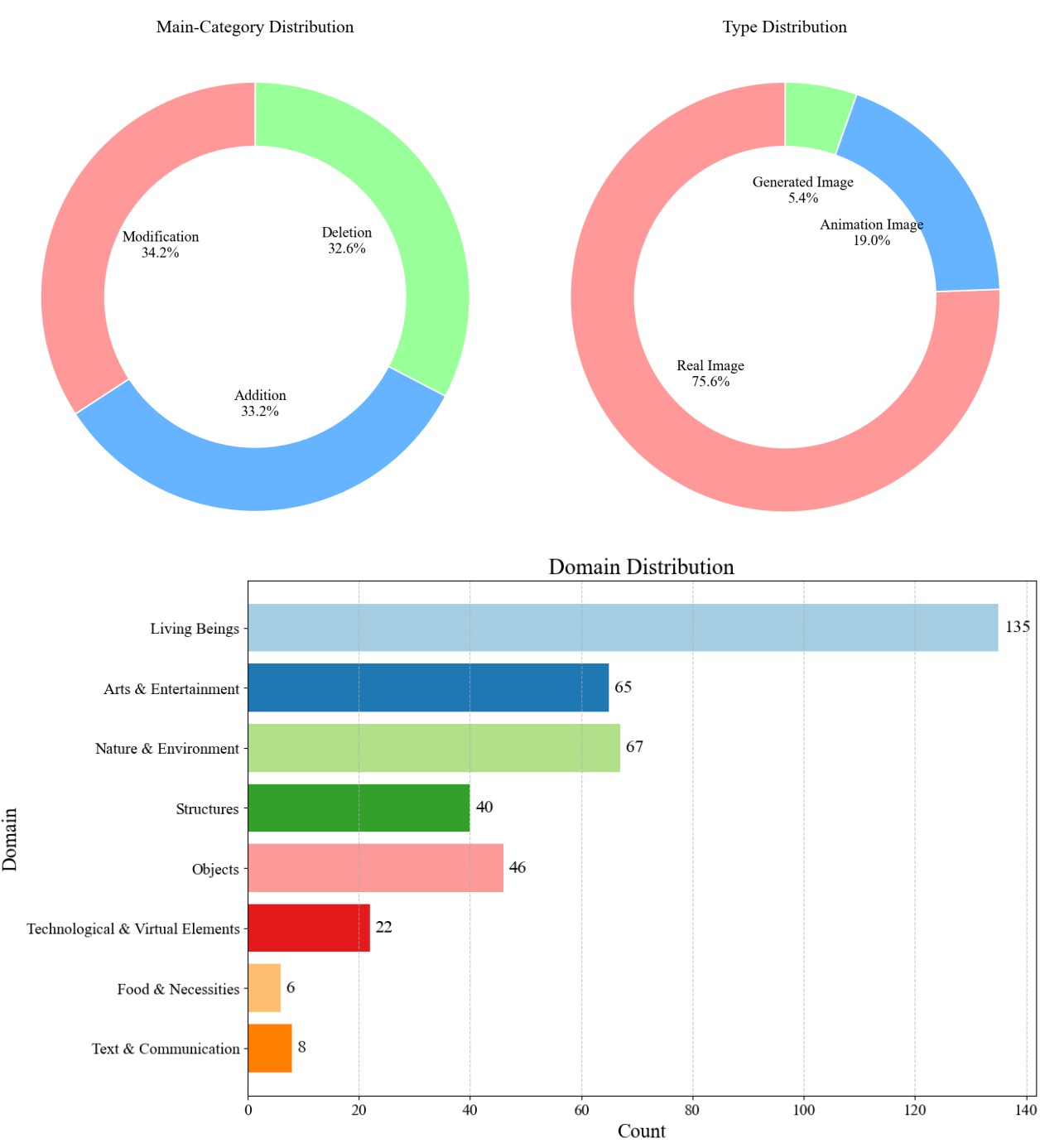

*Figure 4.* Statistical distributions of the OmitI2V benchmark.

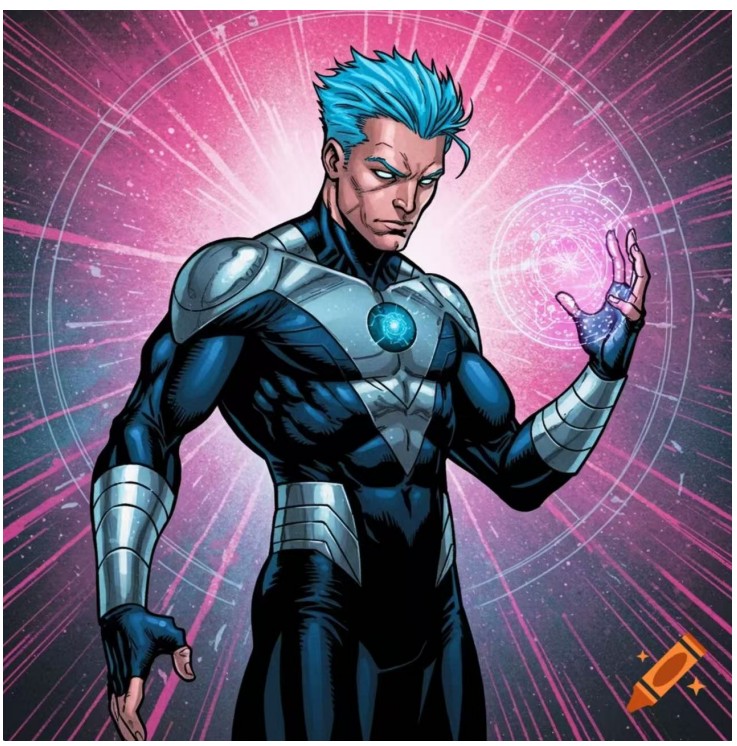

### JSON Sample

**ID:** sample_0
**Image-path:** OmitI2V/modification/pose/human/1.jpg
**Prompt:** The guy pushes out the ball with superpower.
**Expected change:** The man's pose changes to show him pushing the energy orb forward.
**Key:** man pushes an energy ball
**Main Category:** modification
**Sub-category:** pose
**Domain:** human
**Type:** generated image
**Resolution:** 1280x1280
**Aspect Ratio:** 1.0
**Questions:**

1. **Question:** Does the man's pose change to show him pushing forward?
   **Expected Answer:** yes
   **Category:** action correctness

2. **Question:** Does the energy orb move forward as the man pushes it?
   **Expected Answer:** yes
   **Category:** dynamic changes

3. **Question:** Is the man standing still throughout the video?
   **Expected Answer:** no
   **Category:** spatial relationship

*Figure 5.* Sample ("Modification" task) from the OmitI2V benchmark.

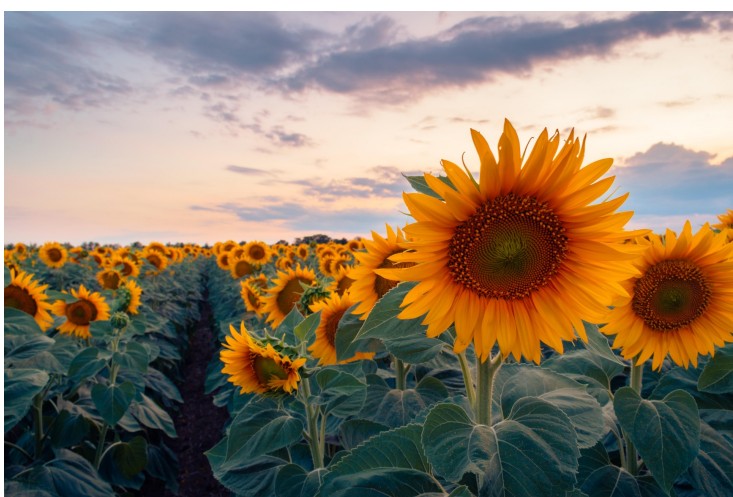

### JSON Sample

**ID:** sample_118
**Image-path:** OmitI2V/modification/style/plant/2.jpg
**Prompt:** The sunflower field gradually shifts into an anime-style rendering, colors becoming more vibrant and outlines turning bold and stylized.
**Expected change:** The realistic sunflowers slowly transform into anime-style flowers with exaggerated textures, bright saturated colors, and defined outlines, while the background remains unchanged.
**Key:** sunflowers to anime style
**Main Category:** modification
**Sub-category:** style
**Domain:** plant
**Type:** real image
**Change:** yes
**Resolution:** 1920x1280
**Aspect Ratio:** 1.5
**Questions:**

1. **Question:** Do the sunflowers change into an anime style?
   **Expected Answer:** yes
   **Category:** dynamic changes

2. **Question:** Are the colors of the sunflowers more vibrant after the transformation?
   **Expected Answer:** yes
   **Category:** attribute accuracy

3. **Question:** Do the outlines of the sunflowers become less defined after the transformation?
   **Expected Answer:** no
   **Category:** attribute accuracy

4. **Question:** Do realistic textures on the sunflowers remain unchanged during the transformation?
   **Expected Answer:** no
   **Category:** attribute accuracy

*Figure 6.* Sample ("Modification" task) from the OmitI2V benchmark.

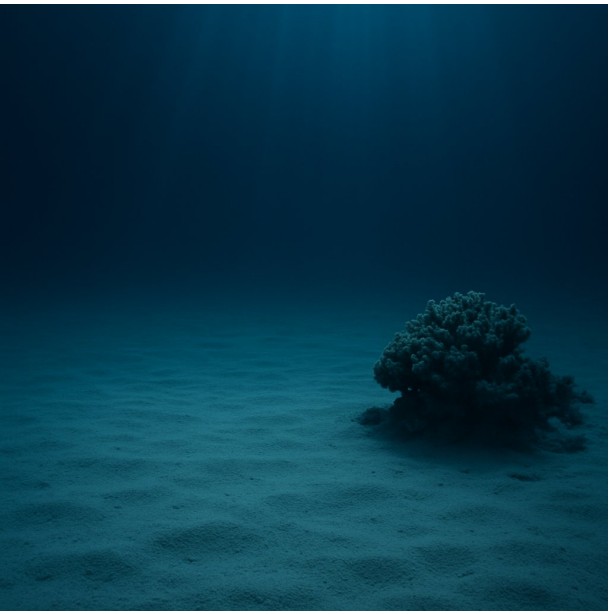

## JSON Sample

**ID:** sample_7
**Image-path:** OmitI2V/addition/appearance/animal/2.png
**Prompt:** Three small fish dart swiftly from the left side of the frame, swimming past vibrant coral and disappearing off to the right.
**Expected change:** Three small fish quickly appear from the left side of the screen and swim through the coral.
**Key:** Three small fish, swim quickly past, past the coral
**Main Category:** addition
**Sub-category:** appearance
**Domain:** animal
**Type:** generated image
**Change:** yes
**Resolution:** 1024x1024
**Aspect Ratio:** 1.0
**Questions:**

1. **Question:** Do the fish swim slowly past the coral?
   **Expected Answer:** no
   **Category:** action correctness

2. **Question:** Is the coral vibrant in color?
   **Expected Answer:** yes
   **Category:** attribute accuracy

3. **Question:** Do the fish appear from the right side of the frame?
   **Expected Answer:** no
   **Category:** spatial relationship

4. **Question:** Do the fish swim past the coral and then disappear off to the right?
   **Expected Answer:** yes
   **Category:** dynamic changes

5. **Question:** Are there more than three small fish in the video?
   **Expected Answer:** no
   **Category:** attribute accuracy

*Figure 7.* Sample ("Addition" task) from the OmitI2V benchmark.

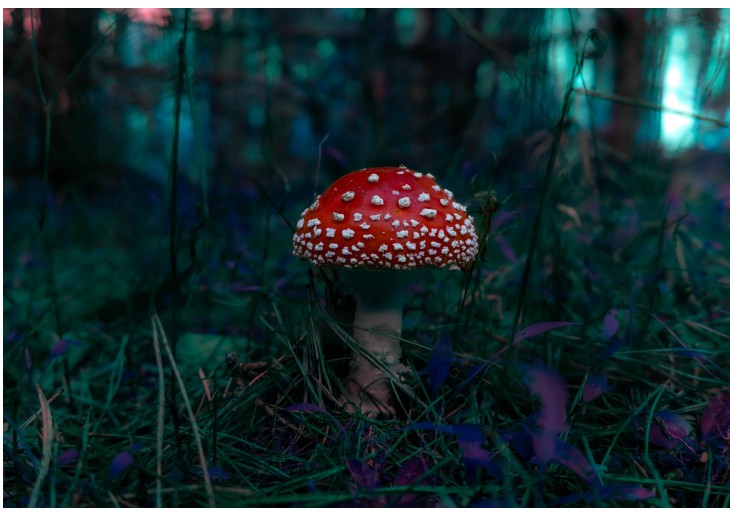

### JSON Sample

**ID:** sample_118
**Image-path:** OmitI2V/addition/object/plant/1.jpg
**Prompt:** A cactus suddenly sprouts and grows tall next to the mushroom, its spines and green stems appearing as it rises from the ground.
**Expected change:** A cactus suddenly sprouts beside the mushroom, growing taller with its spines and green stems clearly forming as it emerges from the ground.
**Key:** cactus grows, next to mushroom
**Main Category:** addition
**Sub-category:** object
**Domain:** plant
**Type:** real image
**Change:** yes
**Resolution:** 1024x1024
**Aspect Ratio:** 1.46
**Questions:**

1. **Question:** Does the cactus grow from the ground up in the video?
   **Expected Answer:** no
   **Category:** action correctness

2. **Question:** Is there a cactus growing next to a tree in the video?
   **Expected Answer:** no
   **Category:** attribute accuracy

3. **Question:** Does the mushroom grow taller than the cactus in the video?
   **Expected Answer:** no
   **Category:** attribute accuracy

4. **Question:** Do multiple cacti grow next to the mushroom in the video?
   **Expected Answer:** yes
   **Category:** object presence

*Figure 8.* Representative sample ("Addition" task) from the OmitI2V benchmark.

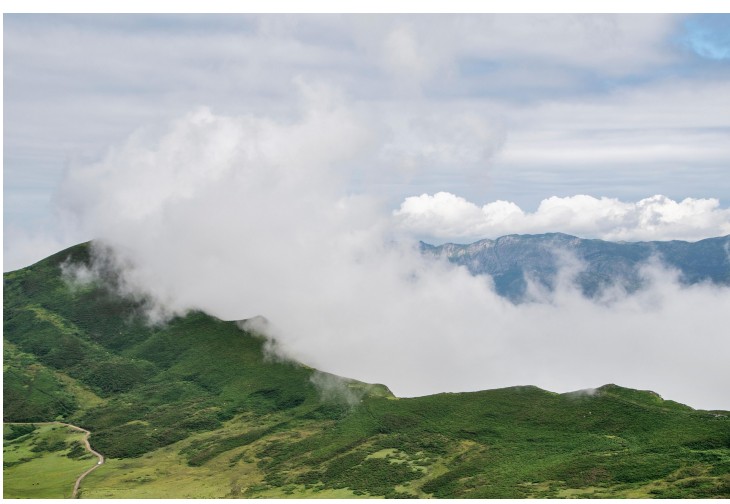

### JSON Sample

**ID:** sample_112
**Image-path:** OmitI2V/deletion/vanish/nature/2.jpg
**Prompt:** The lush green mountain gradually erodes and disappears, leaving behind rolling sand dunes and a barren desert landscape.
**Expected change:** The green vegetation and rocky outcrops of the mountain fade away until only dunes of sand remain.
**Key:** mountain disappears, desert appears
**Main Category:** deletion
**Sub-category:** vanish
**Domain:** nature
**Type:** real image
**Change:** yes
**Resolution:** 1920x1280
**Aspect Ratio:** 1.5
**Questions:**

1. **Question:** Does the mountain remain visible throughout the video?
   **Expected Answer:** no
   **Category:** object presence

2. **Question:** Do sand dunes appear as the mountain erodes?
   **Expected Answer:** yes
   **Category:** action correctness

3. **Question:** Is the landscape at the end of the video primarily composed of lush green vegetation?
   **Expected Answer:** no
   **Category:** attribute accuracy

4. **Question:** Are there any rocky outcrops visible after the mountain has eroded?
   **Expected Answer:** no
   **Category:** object presence

5. **Question:** Does the desert landscape gradually form before the mountain disappears?
   **Expected Answer:** yes
   **Category:** dynamic changes

*Figure 9.* Sample ("Deletion" task) from the OmitI2V benchmark.

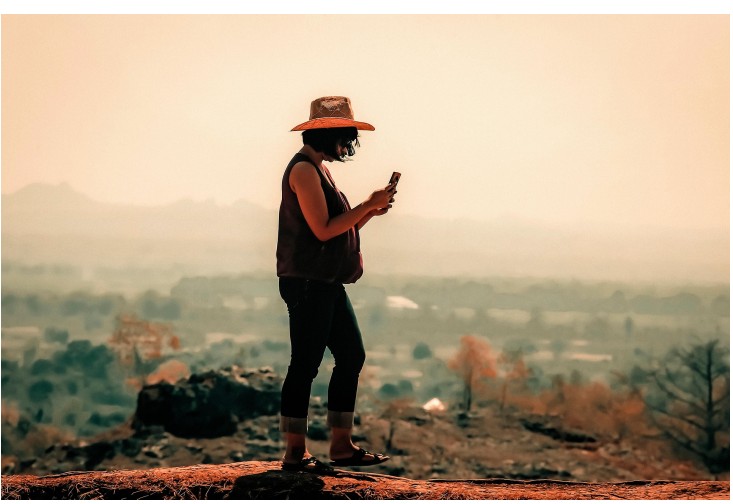

## JSON Sample

**ID:** sample_121
**Image-path:** OmitI2V/deletion/vanish/human/2.jpg
**Prompt:** The woman gradually disappeared as she crouched down
**Expected change:** The woman gradually disappeared as she crouched down
**Key:** duck egg appears
**Main Category:** addition
**Sub-category:** object
**Domain:** human
**Type:** real image
**Resolution:** 1920x1280
**Aspect Ratio:** 1.5
**Questions:**

1. **Question:** Does the woman suddenly disappear?
   **Expected Answer:** no
   **Category:** action correctness

2. **Question:** Does the woman crouch down as she disappears?
   **Expected Answer:** yes
   **Category:** action correctness

3. **Question:** Is a duck egg visible in the scene at any point?
   **Expected Answer:** yes
   **Category:** object presence

4. **Question:** Does the woman's disappearance happen instantly without her crouching down?
   **Expected Answer:** no
   **Category:** dynamic changes

5. **Question:** Does the woman remain fully visible throughout the video?
   **Expected Answer:** no
   **Category:** dynamic changes

*Figure 10.* Sample ("Deletion" task) from the OmitI2V benchmark.

# G. Implementation Details

## G.1. Attention Scaling Modulation.

We implement both *scalar scaling* and *energy-based modulation* within the attention layers. For scalar scaling, a fixed coefficient $\gamma > 1$ is multiplied to either the query or key embeddings. For energy-based modulation, the scaling factor is adaptively computed from the attention logits via a monotonic function, strengthening focus when attention is diffuse.

## G.2. Implementation Details of Block-Level Foreground Analysis

To examine how different transformer blocks distribute their focus between foreground and background, we conduct a block-level study on attention behavior in FramePack.

**Token extraction.** From each self-attention layer, we record the token representations $\mathbf{Z} \in \mathbb{R}^{B \times L \times D}$, where $B$ is the batch size, $L$ is the number of spatio-temporal tokens, and $D$ is the embedding dimension. Tokens are grouped into $T$ segments, each corresponding to a video frame. Frame-level attention scores are then obtained by row-wise summation of the attention matrix.

**Foreground segmentation.** To identify foreground regions, we use the latent noise estimate $\tilde{\epsilon} \in \mathbb{R}^{B \times D \times T \times H \times W}$. We apply PCA (Abdi & Williams, 2010) along the channel axis and retain the top three components, yielding pseudo-RGB projections. These projections are passed into SAM2 (Ravi et al., 2024) to generate binary masks that separate foreground from background.

**Foreground ratio.** Let $\mathbf{M} \in \mathbb{R}^{L \times L}$ denote the attention matrix of a block. For each token $u$, its aggregated attention score is defined as

$$s_u = \frac{1}{L}\sum_{v=1}^{L} M_{uv}. \tag{47}$$

Tokens with $s_u$ larger than a preset threshold are regarded as high-attention tokens. The fraction of these tokens lying inside the foreground mask is defined as the *foreground ratio* $\rho^{(b)}$ for block $b$. A larger $\rho^{(b)}$ implies preference for foreground regions.

We average $\rho^{(b)}$ across 50 diverse prompts to obtain a stable estimate of each block's attention bias. The results are in Table 10.

## G.3. Implementation Details of Step-Level Scheduling

*Step-level scheduling (SGS)* activates modulation only within a predefined interval $[t_{\text{low}}, t_{\text{high}}]$ of the $T$-step denoising trajectory. In experiments, we instantiate three canonical

---

**Algorithm 1:** Selective Scalar Scaling (with BGS and SGS)

**Input:** Query $Q$, Key $K$, Value $V$; scaling factor $\gamma > 1$;
step interval $[t_{\text{low}}, t_{\text{high}}]$; block threshold $\tau$
**Output:** Modulated attention output
**for** *each denoising step $t$* **do**
  **if** $t \in [t_{low}, t_{high}]$ **then**
    // Step-level scheduling
    **for** *each transformer block $l$* **do**
      Compute foreground ratio $r^{(l)}$;
      **if** $r^{(l)} > \tau$ **then**
        // Block-level scheduling
        **if** *modulate Query* **then**
          $Q' \leftarrow \gamma \cdot Q, \quad K' \leftarrow K$;
        **else if** *modulate Key* **then**
          $Q' \leftarrow Q, \quad K' \leftarrow \gamma \cdot K$;
        Compute attention:

$$\text{Attn}^{(l)} = \text{softmax}\left(\frac{Q'(K')^{\top}}{\sqrt{d_k}}\right) V$$

---

windows corresponding to **early**, **middle**, and **late** phases:

$$m_{\text{early}}(t) = \mathbf{1}\left[\tfrac{t}{T} \in [0.00, 0.30]\right],$$
$$m_{\text{middle}}(t) = \mathbf{1}\left[\tfrac{t}{T} \in [0.35, 0.65]\right],$$
$$m_{\text{late}}(t) = \mathbf{1}\left[\tfrac{t}{T} \in [0.70, 1.00]\right].$$

These masks activate ASM over the first 30%, the central 30%, and the final 30% of steps, respectively; the remaining 10% serves as an inactive buffer to avoid boundary artifacts. Unless otherwise noted, we report results for all three schedules and an all-steps variant; based on ablations (Table 5), we adopt the *early-step* schedule as the default.

## G.4. Pseudo-code.

The procedures are summarized in Algorithm 1 and Algorithm 2, which illustrate how block and step level scheduling are combined with scalar scaling or energy-based modulation.

# H. Discussion

## H.1. Exploring the Effect of Different Blur Levels on Generation Results

To better understand the role of image perturbation, we vary the degree of Gaussian blur applied to the input image and analyze its effect on generation quality. As shown in Figure 11, increasing the blur level leads to stronger motion

| Model | Foreground-sensitive blocks |
|---|---|
| FramePack | {0, 2, 4, 5, 6, 7, 10, 11, 12, 13, 14, 15, 16, 17, 18, 19, 20, 21, 23, 25, 26, 27, 31, 33} |
| FramePack F1 | {0, 1, 2, 3, 4, 5, 6, 7, 12, 13, 14, 15, 16, 18, 19, 20, 23, 25, 29, 32, 36, 37, 38, 39} |
| Wan2.1 | {0, 2, 3, 4, 5, 6, 7, 10, 11, 12, 13, 14, 15, 16, 17, 18, 19, 20, 25, 26, 27, 28, 29, 30, 31, 32, 37, 38} |

*Table 10.* **Foreground-sensitive block indices.** We report the blocks identified as foreground-sensitive for FramePack (single-block setting), FramePack-F1, and Wan2.1. These blocks are determined via the foreground ratio analysis described in Section G.2.

---

**Algorithm 2:** Selective Energy-based Modulation (with BGS and SGS)

**Input:** Query $Q$, Key $K$, Value $V$; monotonic function $f(\cdot)$;
step interval $[t_{\text{low}}, t_{\text{high}}]$; block threshold $\tau$
**Output:** Modulated attention output
**for** *each denoising step $t$* **do**
  **if** $t \in [t_{low}, t_{high}]$ **then**
    **for** *each transformer block $l$* **do**
      Compute foreground ratio $r^{(l)}$;
      **if** $r^{(l)} > \tau$ **then**
        Compute logits $z = \frac{QK^{\top}}{\sqrt{d_k}}$;
        Compute adaptive scaling $\gamma = f(z)$ (Equation 10);
        Apply modulation: $K' \leftarrow \gamma \cdot K$ (or $Q'$);
        Compute attention:

$$\text{Attn}^{(l)} = \text{softmax}\left(\frac{Q'(K')^{\top}}{\sqrt{d_k}}\right) V$$

| Method | Semantic Alignment Evaluation | | | ViCLIP Score | | | Visual Quality Evaluation | |
|---|---|---|---|---|---|---|---|---|
| | Modification ↑ | Addition ↑ | Deletion ↑ | Modification ↑ | Addition ↑ | Deletion ↑ | Dynamic Degree ↑ | Aesthetic Quality ↑ |
| *FramePack* | | | | | | | | |
| Original | 64.99 | 68.55 | 58.14 | 20.83 | 21.08 | 20.43 | 20.05 | 63.94 |
| Scalar scaling $\gamma = 1.25$ | 66.97 | 71.91 | 59.52 | 21.00 | 21.74 | 20.78 | 28.02 | 63.67 |
| Scalar scaling $\gamma = 1.35$ | 67.15 | 73.44 | 59.86 | 21.38 | 22.03 | 21.05 | 28.28 | 63.41 |
| CAS (adaptive) | 66.79 | 72.37 | 59.01 | – | – | – | 26.79 | 63.77 |
| *FramePack F1* | | | | | | | | |
| Original | 64.45 | 67.79 | 58.50 | 21.06 | 19.91 | 20.61 | 24.42 | 63.10 |
| Scalar scaling $\gamma = 1.25$ | 68.04 | 70.21 | 60.12 | 21.75 | 20.57 | 20.92 | 32.68 | 62.12 |
| Scalar scaling $\gamma = 1.35$ | 70.02 | 71.45 | 61.06 | 21.78 | 21.04 | 20.99 | 33.16 | 62.11 |

*Table 11.* Ablation on the scaling coefficient. *CAS (adaptive)* is the per-token Conflict-Aware Scaling variant (Appendix H.3), which sets $\gamma$ automatically from the image/text attention conflict; ViCLIP scores for this variant are not measured ("–").

trol without overly sacrificing the overall aesthetics of the generated video.

**Transferability of the scaling factor.** This monotonic behavior is not incidental but follows directly from our analysis: Lemma 2 (entropy monotonicity under scaling, Appendix B.2) shows that increasing the scaling factor $\gamma$ monotonically reduces the conditioning entropy, so the semantic–quality trade-off is predictable by design. Table 11 confirms this empirically on both backbones: larger $\gamma$ monotonically improves semantic fidelity while aesthetic quality decreases only gradually. As a result, $\gamma=1.35$ transfers across backbones without backbone-specific search, analogous to selecting the guidance strength in classifier-free guidance.

### H.3. Conflict-Aware Adaptive Scaling (CAS)

The fixed scaling factor $\gamma$ applies the same strength to every token. A natural question is whether $\gamma$ can instead be set automatically per token, concentrating modulation only where the image and text conditions actually compete. To this end, we study a fully automated, per-token adaptive variant, termed *Conflict-Aware Scaling* (CAS), which requires no manual tuning:

$$C = \frac{\max(A_{\text{img}} - A_{\text{text}}, 0)}{A_{\text{img}} + A_{\text{text}} + \epsilon} \cdot H, \qquad \gamma = 1 + C \quad (0 \le C \le 0.35),$$
(48)

where $A_{\text{img}}$ and $A_{\text{text}}$ denote the attention mass placed on image and text tokens, respectively, and $H$ is the attention entropy. Intuitively, $C$ is large only when attention is both dominated by the image condition (large $A_{\text{img}} - A_{\text{text}}$) and diffuse (large $H$) — precisely the configurations in which the text prompt is at risk of being neglected — while the clamp keeps $\gamma$ within $[1, 1.35]$, recovering the fixed setting as its upper bound.

and more complex subject dynamics, but at the cost of degraded visual fidelity. Conversely, mild blur provides a balanced improvement, enhancing semantic alignment while largely preserving perceptual quality. This highlights a trade-off between motion richness and aesthetic sharpness, suggesting that blur can be interpreted as a controllable proxy for motion strength.

### H.2. Ablation on the Scaling Coefficient

We also conduct ablation studies on the effect of the scaling coefficient applied in our guidance mechanism. The quantitative results are summarized in Figure 11. We observe a clear trend: as the scaling coefficient increases, both semantic fidelity and dynamic degree consistently improve, indicating stronger alignment with the conditioning signal. However, this improvement comes at the cost of aesthetic quality, which degrades as the coefficient grows. This trade-off highlights the importance of choosing a moderate coefficient that balances semantic consistency with visual appeal. In practice, we select a coefficient that achieves a satisfactory compromise, ensuring faithful semantic con-

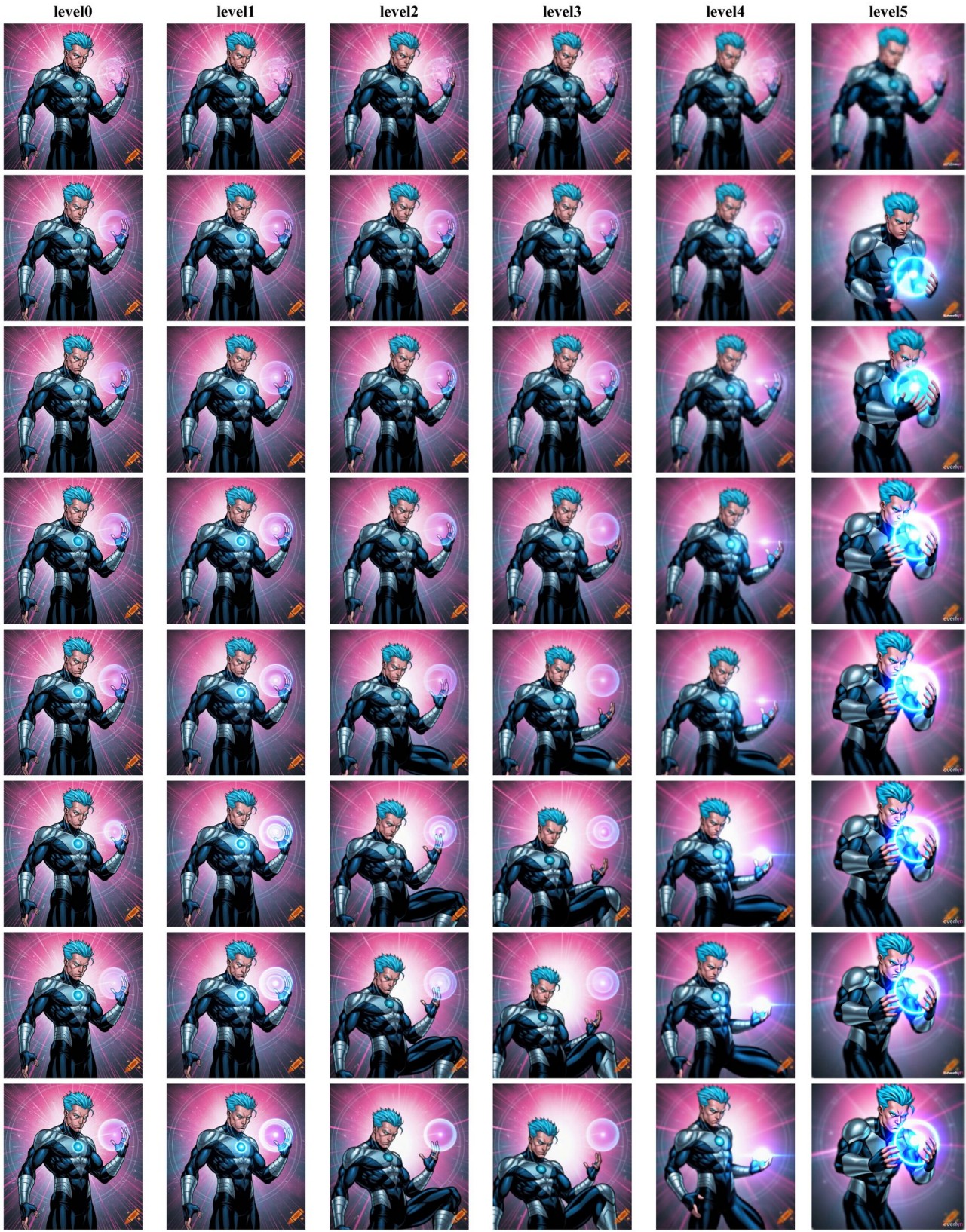

*Figure 11.* Effect of varying Gaussian blur levels on I2V generation. Higher blur increases motion amplitude and subject complexity, but reduces visual fidelity. Mild blur improves semantic alignment while largely preserving perceptual quality.

| Model | Runtime per video (s) ↓ | | Overhead (%) ↓ |
|---|---|---|---|
| | Original | +Scalar | |
| FramePack | 129.90 | 130.02 | +0.09 |
| FramePack F1 | 117.05 | 117.08 | +0.03 |
| Wan2.1 | 445.66 | 445.71 | +0.01 |

*Table 12.* **Inference time comparison.** Our method introduces **negligible** inference overhead. Overhead is computed as $\frac{\text{Method}-\text{Original}}{\text{Original}} \times 100\%$. Experiments are conducted on a single NVIDIA H100 (80 GB). FramePack and FramePack-F1 generate $832\times480$ videos with 177 frames. Wan2.1 generates $800\times480$ videos with 81 frames.

As reported in Table 11, CAS approaches fixed scaling on semantic fidelity (e.g. Addition 72.37 vs. 73.44) while better preserving aesthetic quality (63.77 vs. 63.41). This confirms that conflict-driven, per-token adaptation is an effective, search-free alternative to a globally tuned coefficient, modulating only the tokens where image–text conflict is highest.

### H.4. Inference Efficiency

Our method selectively modulates attention only at foreground-sensitive blocks and within a limited interval of denoising steps; it is important to understand the impact on computational cost.

Let $L$ denote the total number of transformer blocks and $T$ the number of denoising steps. Suppose attention modulation is applied to $L_s \leq L$ blocks over $T_s \leq T$ steps. Then, the additional attention computation introduced by our scaling mechanism can be approximated as:

$$\Delta\text{FLOPs} \approx \frac{L_s}{L} \cdot \frac{T_s}{T} \cdot \text{FLOPs}_{\text{attn}}, \qquad (49)$$

where $\text{FLOPs}_{\text{attn}}$ denotes the cost of a single attention operation in one block. This expression indicates that, by restricting modulation to a subset of blocks and steps, the computational overhead remains a small fraction of the total generation cost.

Empirically, as shown in Table 12, our method introduces only **negligible** inference overhead.

### H.5. Results on Other I2V Benchmarks

To further validate the generalizability of our approach, we conduct experiments on the VBenchI2V benchmark. The results, summarized in Tables 13 - 14, show that our method consistently achieves higher average quality scores compared to the baselines. While the overall I2V score remains comparable to the baseline methods.

More specifically, when the scale coefficient is set below 1,

| Metric | FramePack | | | | |
|---|---|---|---|---|---|
| | Original | $\gamma=0.95$ | $\gamma=1.15$ | $\gamma=1.25$ | $\gamma=1.35$ |
| *Video-Condition Dimension* | | | | | |
| I2V subject | 98.89 | **98.93** | 98.80 | 98.74 | 98.70 |
| I2V background | 99.01 | 99.02 | 98.96 | **99.15** | 99.01 |
| Camera motion | 61.21 | 60.81 | **61.73** | 61.47 | 60.63 |
| Average I2V score | 86.37 | 86.25 | **86.50** | 86.45 | 86.11 |
| *Video-Quality Dimension* | | | | | |
| Subject consistency | 96.53 | **96.65** | 96.13 | 95.89 | 95.58 |
| Background consistency | 97.88 | 97.83 | 97.78 | **98.22** | 97.75 |
| Motion smoothness | 99.53 | **99.54** | 99.50 | 99.48 | 99.45 |
| Dynamic degree | 28.86 | 26.02 | 32.93 | 35.77 | **38.61** |
| Aesthetic quality | 61.71 | 61.62 | **61.74** | 61.19 | 61.40 |
| Imaging quality | 70.62 | **70.70** | 70.44 | 70.23 | 70.45 |
| Average quality score | 75.86 | 75.39 | 76.42 | 76.80 | **77.21** |

*Table 13.* Ablation about scaling coefficient (Transposed).

| Metric | FramePack F1 | | | | |
|---|---|---|---|---|---|
| | Original | $\gamma=0.95$ | $\gamma=1.15$ | $\gamma=1.25$ | $\gamma=1.35$ |
| *Video-Condition Dimension* | | | | | |
| I2V subject | 98.88 | **98.91** | 98.68 | 98.63 | 98.58 |
| I2V background | 99.18 | **99.19** | 99.12 | 99.08 | 99.06 |
| Camera motion | 49.54 | **49.93** | 48.75 | 48.23 | 49.80 |
| Average I2V score | 82.53 | **82.68** | 82.18 | 81.98 | 82.48 |
| *Video-Quality Dimension* | | | | | |
| Subject consistency | 94.94 | **95.16** | 94.33 | 94.03 | 93.95 |
| Background consistency | 97.40 | **97.46** | 97.20 | 97.10 | 96.98 |
| Motion smoothness | 99.40 | **99.42** | 99.36 | 99.33 | 99.31 |
| Dynamic degree | 33.33 | 30.89 | 40.65 | 42.68 | **43.90** |
| Aesthetic quality | 61.25 | **61.29** | 61.10 | 61.06 | 60.97 |
| Imaging quality | 70.28 | **70.31** | 70.04 | 70.03 | 69.90 |
| Average quality score | 76.10 | 75.76 | 77.11 | 77.37 | **77.50** |

*Table 14.* Ablation about scaling coefficient (Transposed).

all metrics except *Dynamic Degree* improve over the baseline. In contrast, when the coefficient is greater than 1, the *Dynamic Degree* metric increases significantly, while other indicators remain within a stable range. **This is analogous to the temperature parameter in large language models: by simply adjusting a single scale value, users can flexibly balance between aesthetic quality (smaller scale) and prompt fidelity (larger scale).**

These results highlight the simplicity and effectiveness of our method. Without introducing additional training or complex modules, our approach provides a lightweight method for controlling video generation quality across diverse I2V benchmarks.

### H.6. Validating Semantic Fidelity Metrics with Human Evaluation

To validate the effectiveness of our metrics, we conduct a user study on a total of 60 samples, sampling 20 instances per semantic change type (*addition*, *deletion*, *modification*) with 5 people.

**Setup.** For each sample, we form a triplet: the original prompt and image, the video generated by a baseline model, and the video generated using our method. Human annotators rated each video along two dimensions: *semantic*

| Method | Semantic Fidelity | | | Aesthetic Quality | | |
|---|---|---|---|---|---|---|
| | Addition | Deletion | Modification | Addition | Deletion | Modification |
| Framepack | 3.05 | 3.20 | 3.16 | 5.70 | 5.60 | 5.65 |
| Framepack + Ours | 5.72 | 5.80 | 5.82 | 5.63 | 5.56 | 5.63 |

*Table 15.* Human ratings (1–7 scale) for each semantic change type. Our metrics correlate well with human judgment across addition, deletion, and modification.

*fidelity* and *aesthetic quality*, using a 1–7 Likert scale.

**Results.** Table 15 summarizes the average human scores compared with our OmitI2V metrics. We observe that the human ratings consistently align with the metric trends: videos generated with our method achieve higher semantic fidelity while maintaining comparable aesthetic quality.

### H.7. Analysis of the VQA-based Semantic Evaluator on OmitI2V

Our main semantic fidelity metric on OmitI2V is derived from a VQA model (Qwen2.5-VL-32B) answering yes/no questions about whether the requested edit has been correctly executed. Since this introduces a potential source of bias, we explicitly quantify its reliability and inspect its typical failure modes.

**Quantitative error analysis.** We manually annotated OmitI2V samples generated by FramePack V1 and computed False Positive (FP) and False Negative (FN) statistics for each edit type. Table 16 summarizes the error rates:

| Main category | FP rate | FN rate | Overall error |
|---|---|---|---|
| Addition | 0.78% | 1.94% | 2.71% |
| Deletion | 0.92% | 3.15% | 4.07% |
| Modification | 0.63% | 2.76% | 3.38% |
| All | 0.77% | 2.61% | 3.38% |

*Table 16.* Error statistics of the Qwen2.5-VL-32B evaluator on OmitI2V (FramePack V1). We report the FP rate, the FN rate, and overall error for each edit type.

The overall error remains around 3–4% across all three edit types, indicating that Qwen2.5-VL-32B is generally reliable as an automatic evaluator on this benchmark.

**Observed systematic tendencies.** When inspecting the incorrect cases, we observe two mild but interpretable tendencies:

- **False negatives on small or partially occluded objects (conservative behavior).** In some *addition* and *deletion* clips, the evaluator answers "no" to object-presence questions even though the target object is present but small, partially occluded, or overshadowed by a larger foreground object. A typical pattern is:

  > *Question:* "Is a cat visible in the video?"
  > *Ground truth:* Yes
  > *Model answer:* No, the video shows a bear walking through a valley, not a cat.

  Here, the cat is indeed visible, but the evaluator attends mainly to the dominant animal and misses the smaller one, leading to a conservative negative prediction.

- **Over-endorsement of the prompt effect (slight positive bias).** In a few *modification* clips, the evaluator correctly detects that some visual change occurs, but overstates the strength of the edit. For example:

  > *Question:* "Does the instructor fade out of view while still holding the beaker?"
  > *Ground truth:* No
  > *Model answer (abridged):* Yes, the instructor gradually fades out of view while still holding the beaker, indicating that their presence is being removed from the scene.

  Our frame-level inspection shows only mild transparency/compositing changes rather than a full fade-out. In such cases, the evaluator captures a real change but hallucinates a stronger, cleaner effect than what is actually rendered.

The identified failure cases mostly involve borderline or subtle situations (tiny objects, very mild appearance changes), whereas the majority of OmitI2V edits are clear semantic operations (adding, removing, or modifying an object), for which the evaluator behaves consistently.

## I. Qualitative Visualization of Evaluation Results

To further demonstrate the effectiveness of our method, we provide qualitative visualizations comparing the original videos, baseline methods (*Framepack*, *Framepack F1*, *Wan2.1*), and our approach. These comparisons highlight improvements in both semantic consistency and visual quality, showing that our method produces more faithful renderings with better alignment to the input prompts. Representative examples are presented in Figure 12 to Figure 17.

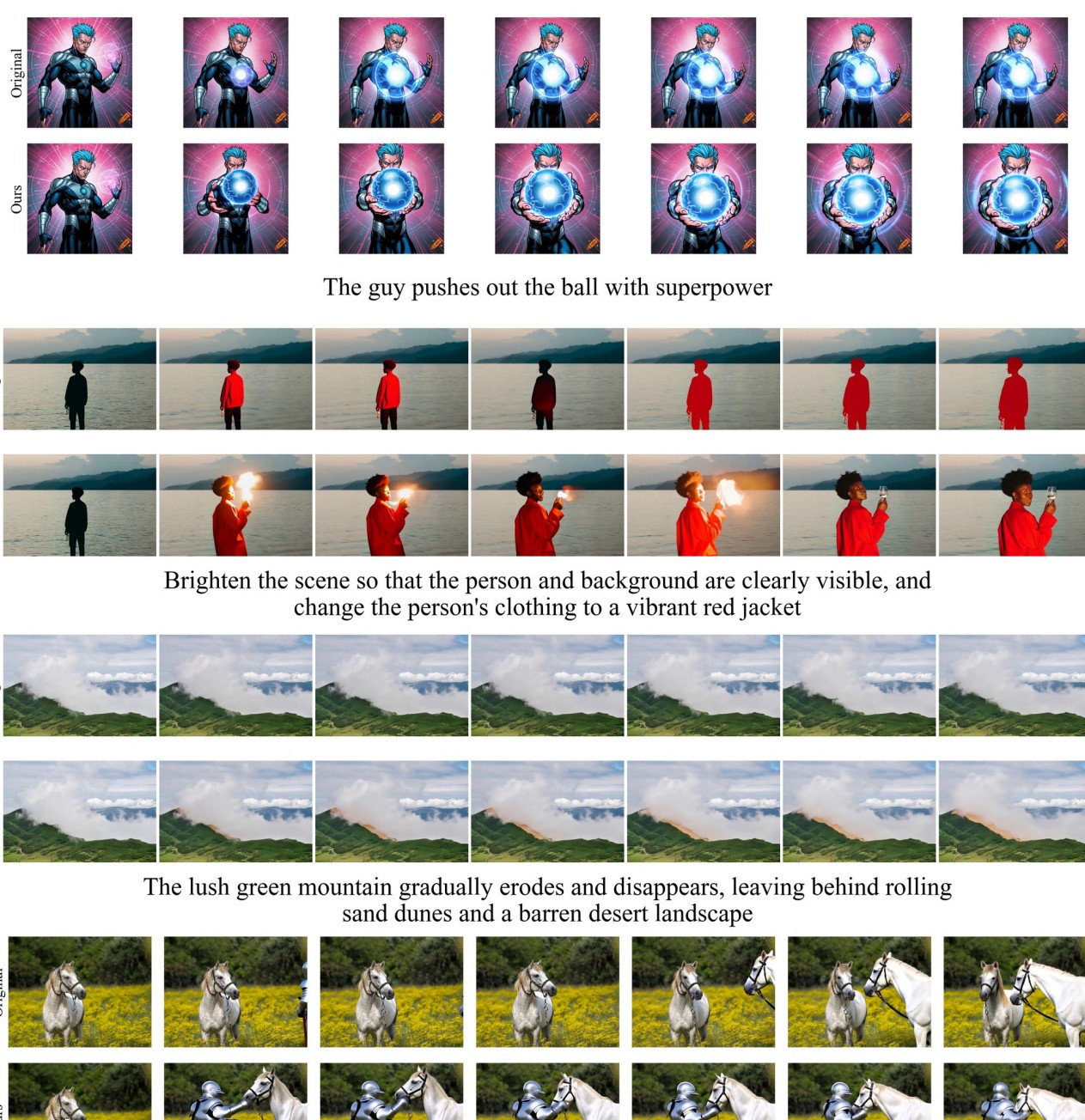

*Figure 12.* Example comparison of our method and *Framepack*.

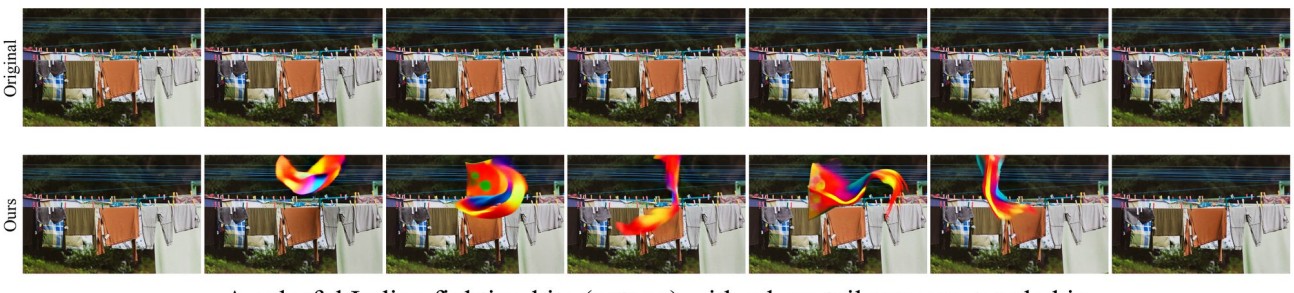

A colorful Indian fighting kite (patang) with a long tail appears, tangled in
the clothesline

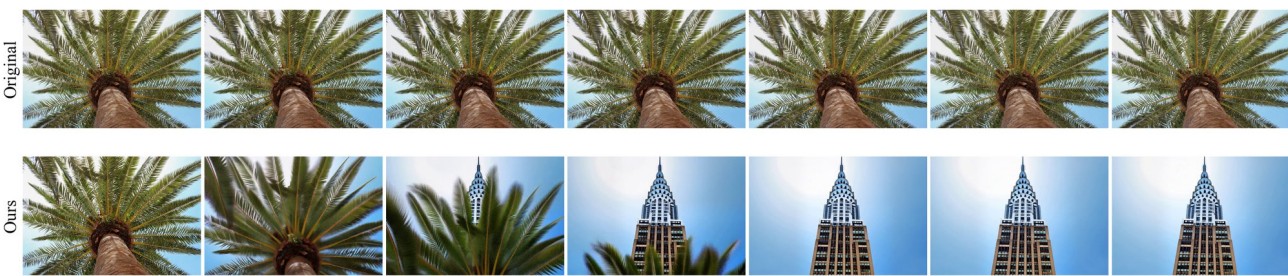

A classic Art Deco skyscraper, reminiscent of New York's Chrysler Building,
appears rising into the sky behind the palm fronds

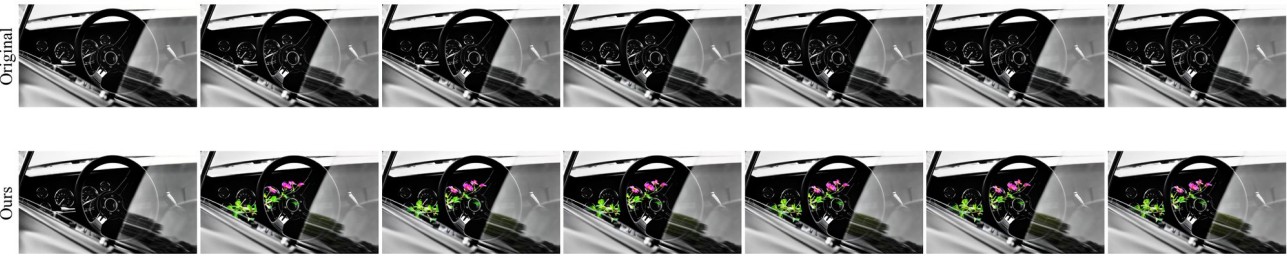

Flowers grew out of the steering wheel, and the picture became colorful

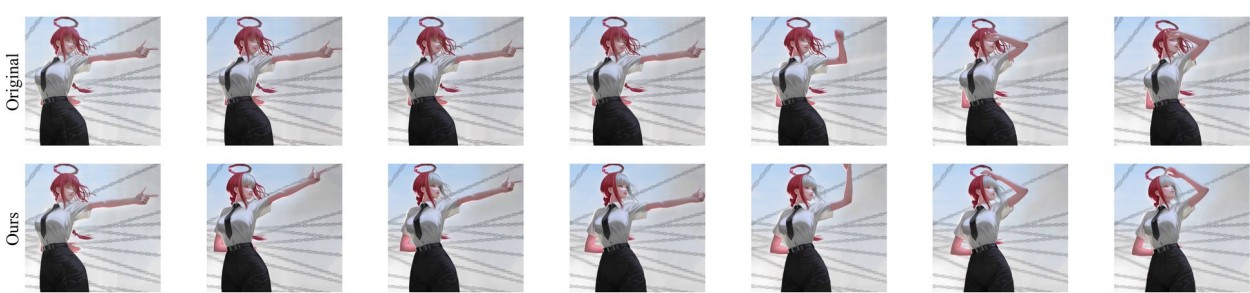

The character's hair color gradually shift from red to white, transitioning
smoothly over time

*Figure 13.* Example comparison of our method and *Framepack*.

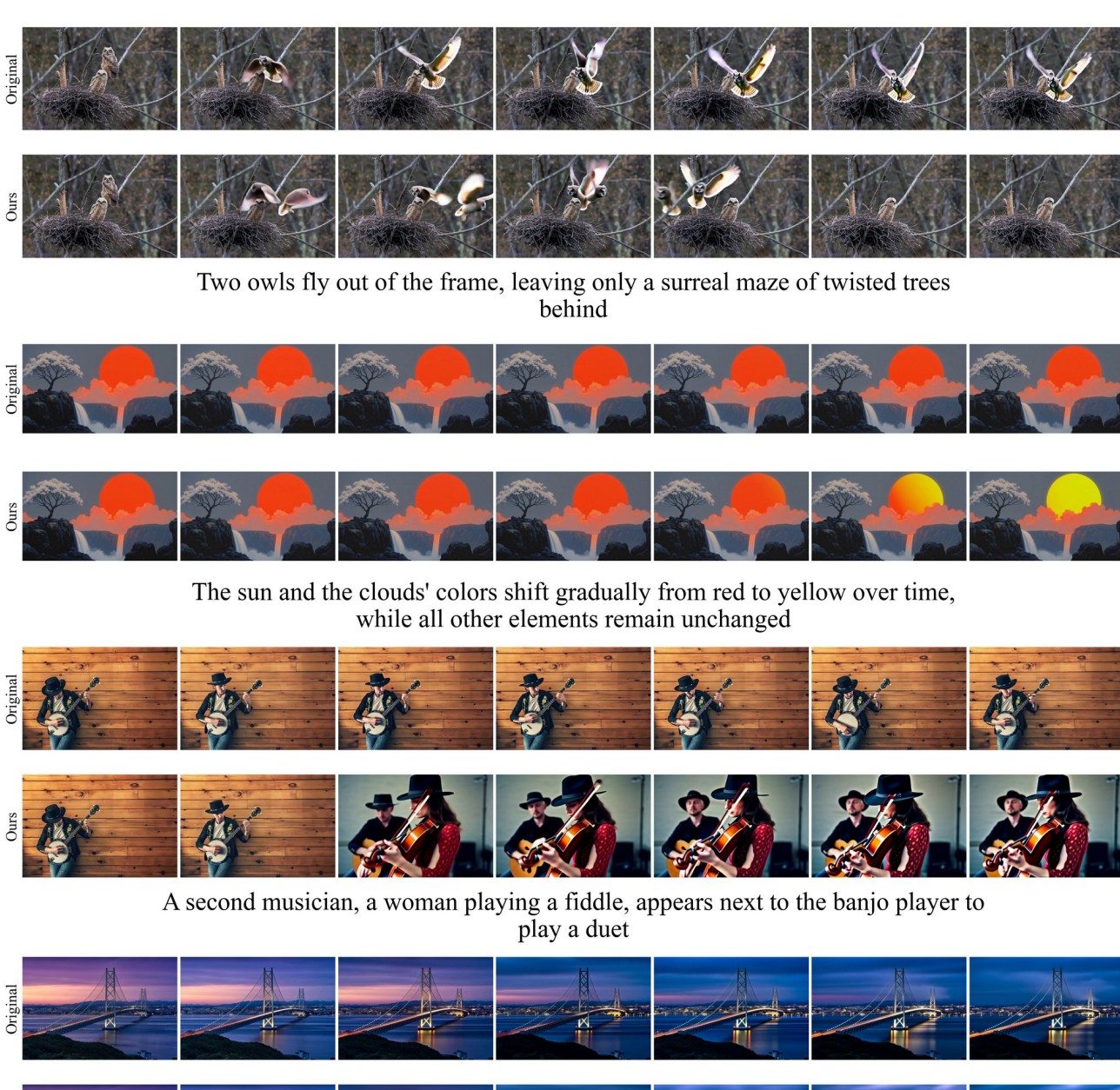

*Figure 14.* Example comparison of our method and *Framepack F1*.

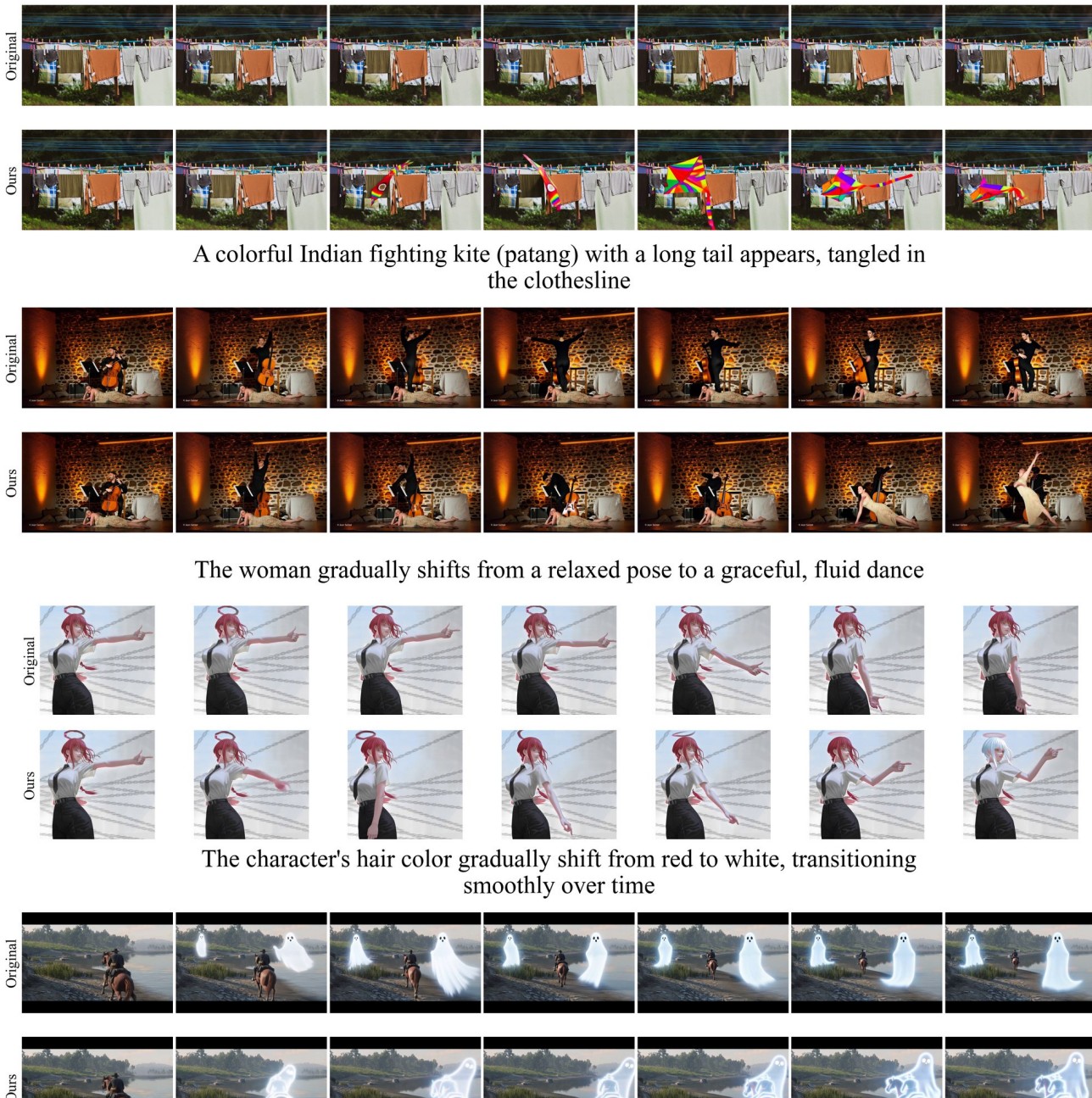

A colorful Indian fighting kite (patang) with a long tail appears, tangled in the clothesline

The woman gradually shifts from a relaxed pose to a graceful, fluid dance

The character's hair color gradually shift from red to white, transitioning smoothly over time

The man on horseback gradually transforms into a ghostly silhouette, his form slowly fading and dissolving into translucent light, while his horse begins to fade into the serene landscape

*Figure 15.* Example comparison of our method and *Framepack F1*.

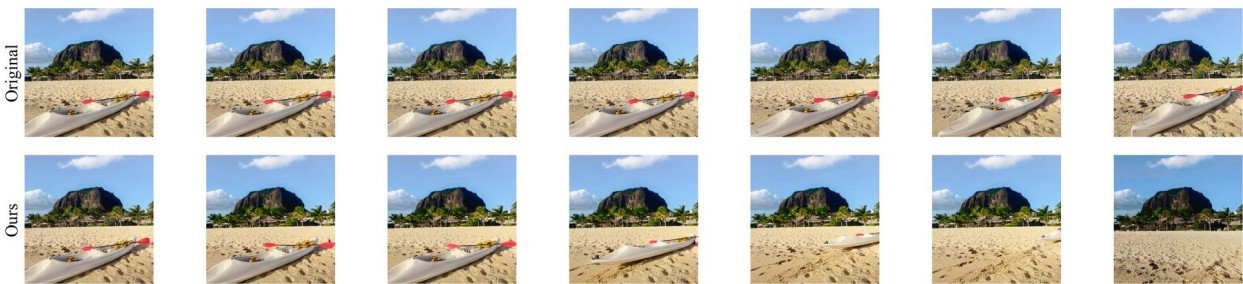

The kayak quietly powers itself forward until it disappears from view

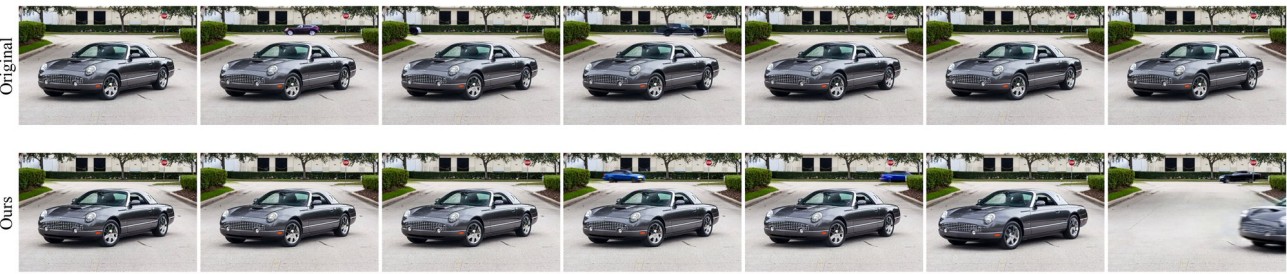

The gray convertible car move forward and gradually drives out of sight

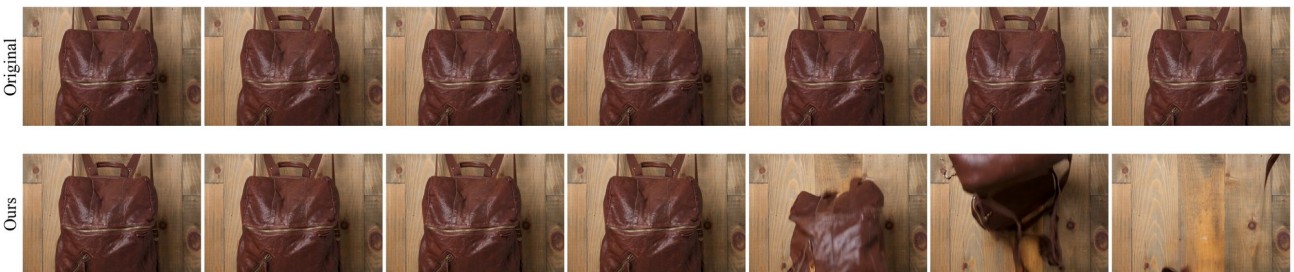

The backpack falls out of the frame

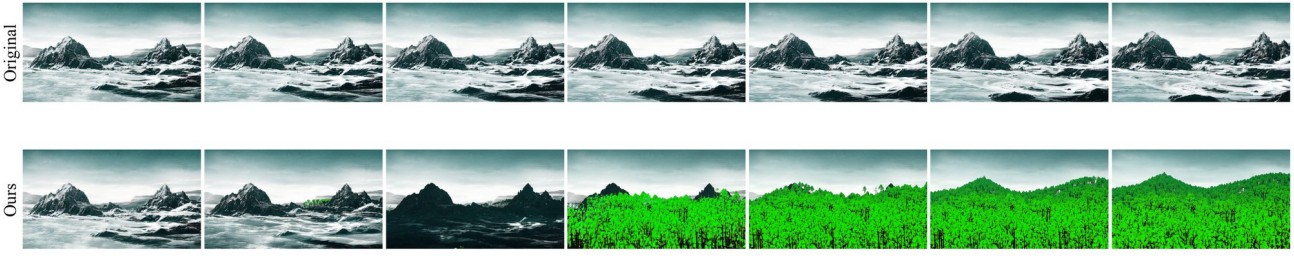

Trees sprout and quickly grow across the mountains, covering the rocky slopes with lush green foliage

*Figure 16.* Example comparison of our method and *Wan2.1*.

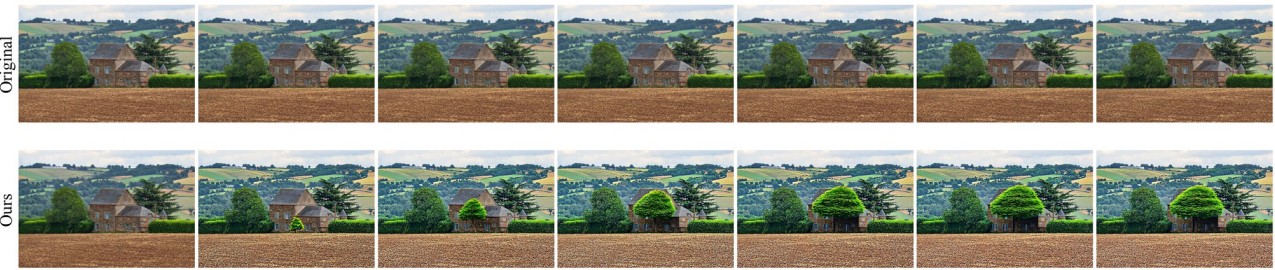

A young tree sprouts at the front of the house and grows quickly until it stands full-height in front of the facade

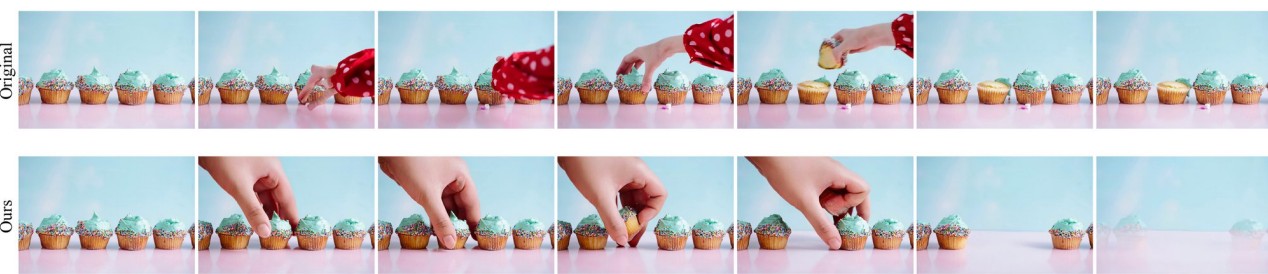

A hand reaches into the frame, picks up one cupcake from the row, and removes it until it vanishes from view

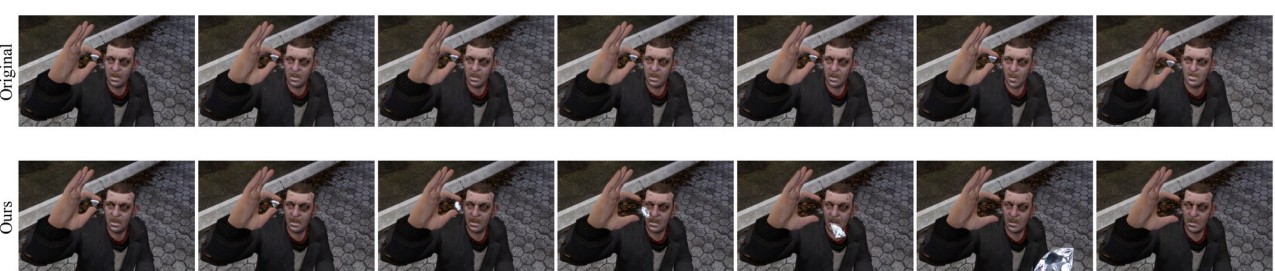

The man extending his arm forward and releasing the diamond from his fingers

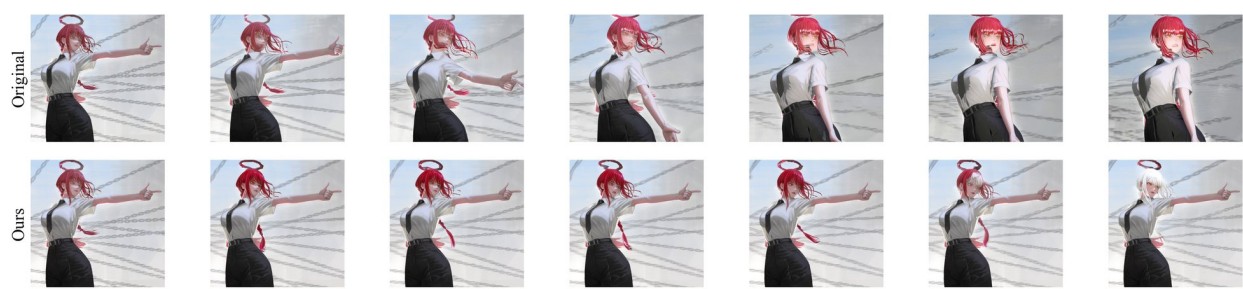

The character's hair color gradually shift from red to white, transitioning smoothly over time

*Figure 17.* Example comparison of our method and *Wan2.1.*

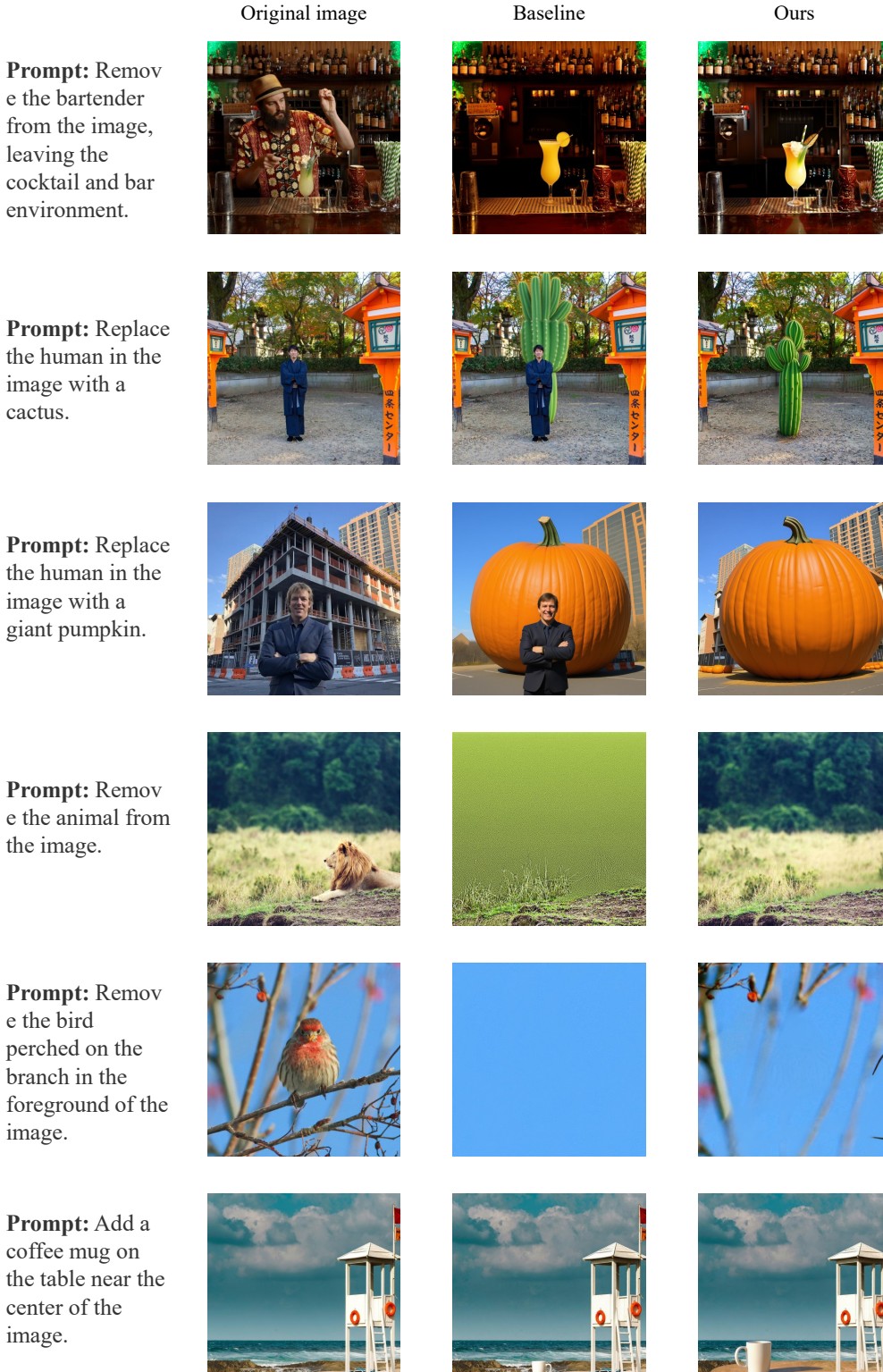

*Figure 18.* Example comparison of our method and baseline on Imgedit benchmark.

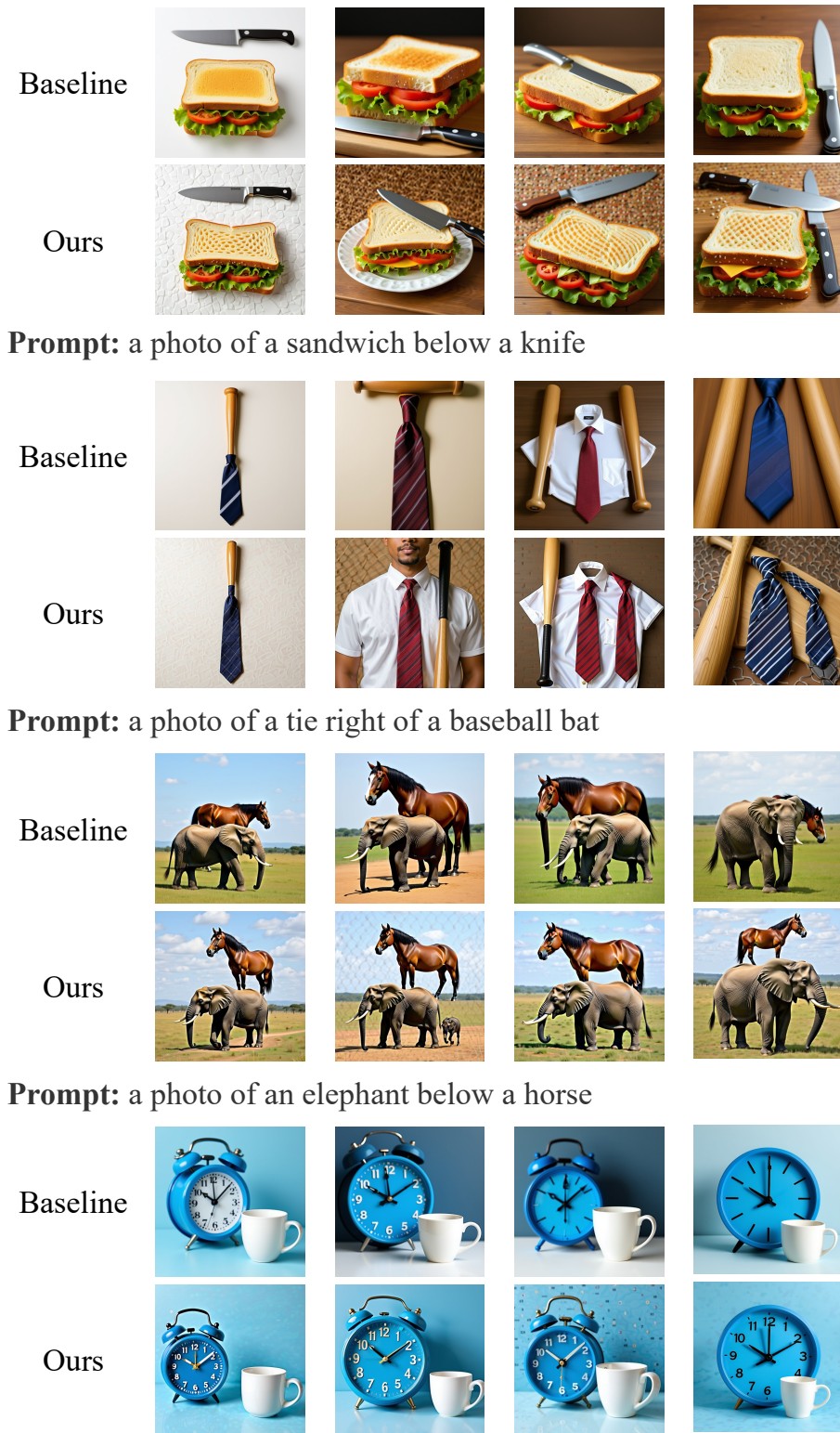

**Prompt:** a photo of a sandwich below a knife

**Prompt:** a photo of a tie right of a baseball bat

**Prompt:** a photo of an elephant below a horse

**Prompt:** a photo of a blue clock and a white cup

*Figure 19.* Example comparison of our method and baseline on Geneval benchmark.

Prompt: a hair drier on the right of a toothbrush front view

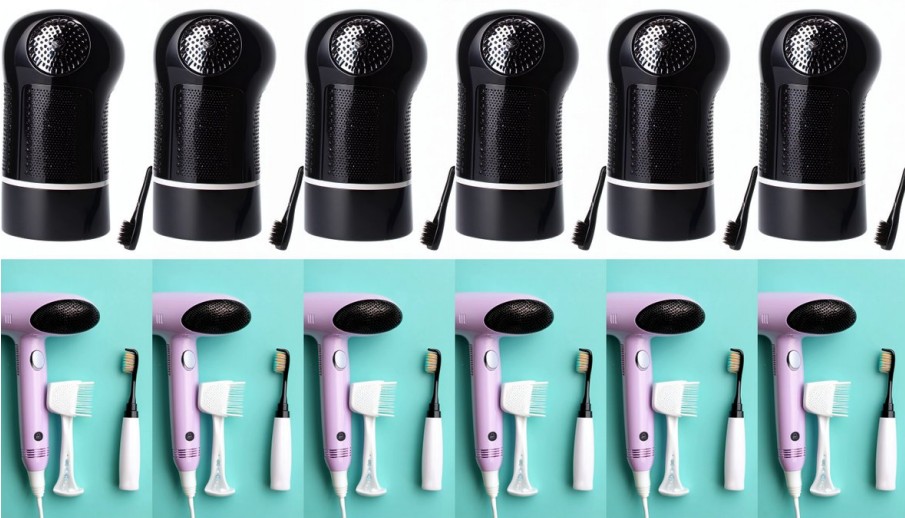

Prompt: a handbag and a tie

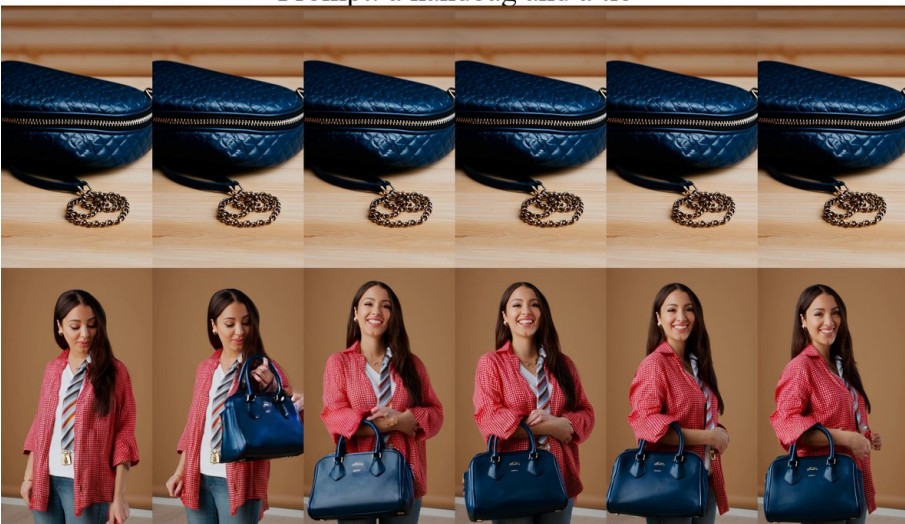

Prompt: a tie and a suitcase

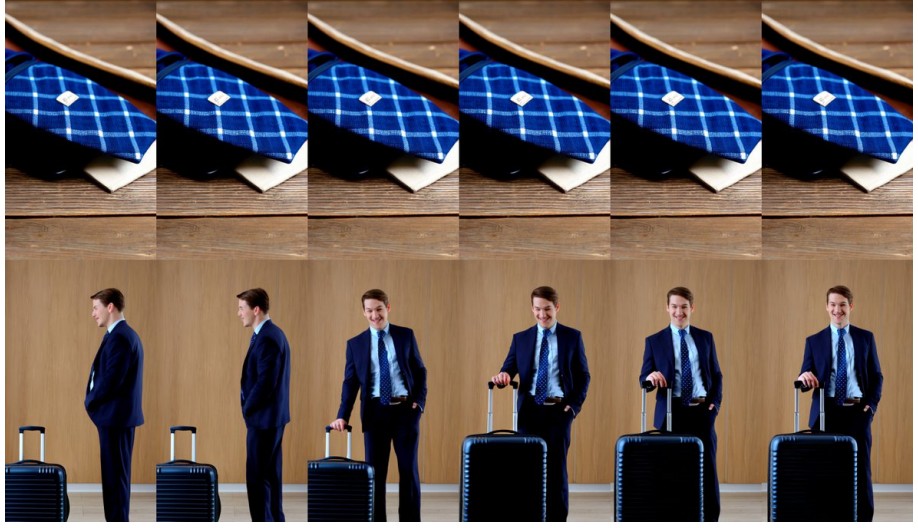

*Figure 20.* Example comparison of our method and Wan2.1-T2V-1.3B.

Prompt: an orange on the top of a carrot front view

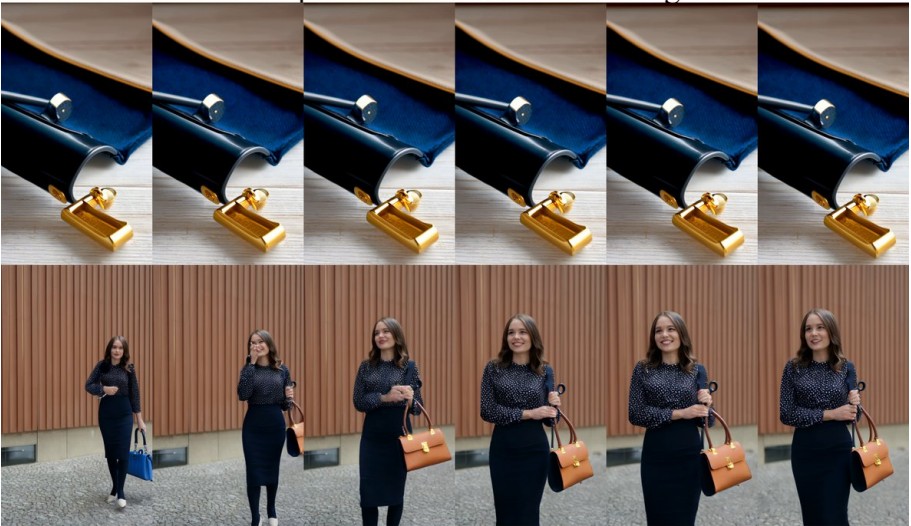

Prompt: an umbrella and a handbag

Prompt: skis on the bottom of a snowboard front view

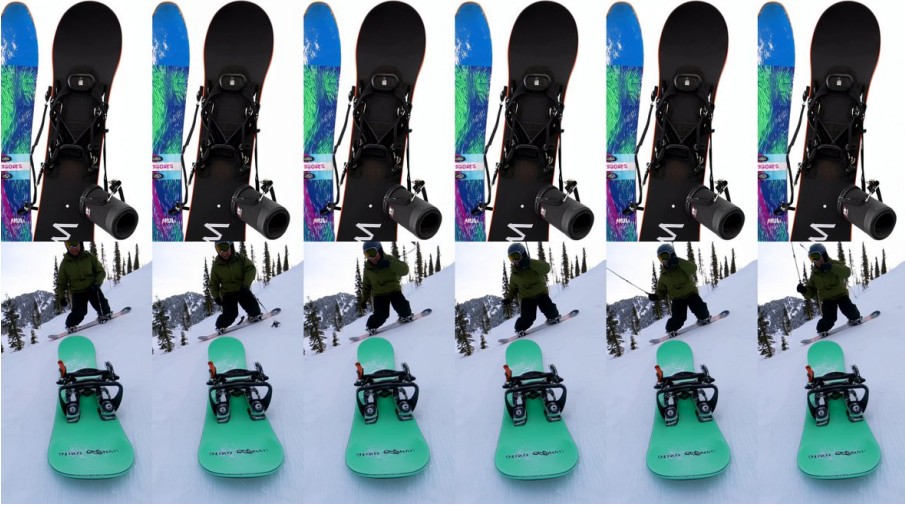

*Figure 21.* Example comparison of our method and Wan2.1-T2V-1.3B.

