# OpenReview forum: "AlignVid: Taming Visual Dominance via Training-Free Attention Modulation in Text-guided Image-to-Video Generation"
_ICML.cc/2026/Conference — ICML 2026 regular_

### Official Review · Reviewer_ZZNs · 2026-03-09

**Soundness:** 3
**Presentation:** 3
**Significance:** 3
**Originality:** 3
**Overall Recommendation:** 4
**Confidence:** 3

**Summary:**

The paper's notable concept concerns the semantic negligence problem in text-guided image-to-video generation, where models fail to execute substantial prompt-driven edits due to visual dominance. The authors discover through a pilot study that blurring the reference image reduces attention entropy and restores the priority of text prompts. Overall, this manuscript's key theme is the introduction of AlignVid, a training-free framework designed to recalibrate the internal attention distribution of pretrained models. This framework utilizes Attention Scaling Modulation to mathematically sharpen the attention focus and reduce entropy, paired with Guidance Scheduling to maintain generation stability. Furthermore, the authors propose the OmitI2V benchmark to evaluate prompt adherence across various editing scenarios.

**Compliance With Llm Reviewing Policy:**

Affirmed.

**Final Justification:**

The author resolved my doubts, and I'm willing to give a higher rating.

**Key Questions For Authors:**

1. Theoretical Alignment of ASM and Blurring:
How does the mathematical sharpening of ASM align with the physical blurring in the pilot study? Blurring suppresses visual redundancy, whereas ASM amplifies existing salient features. How does the method prevent amplifying strong visual distractors? A clear theoretical or empirical justification would significantly strengthen the soundness of the paper.

2. Generalization of Guidance Scheduling:
Is the Guidance Scheduling strategy universally applicable, or is it heuristically tuned for the specific base model evaluated? How sensitive is the method to scheduling hyperparameters across different backbone models?

3. Full Video Quality and Temporal Coherence:
Can the authors provide full video results (not just static frames) to demonstrate temporal coherence and motion quality? Specifically, does ASM introduce any temporal flickering or instability, and how is this addressed?

4. User Study:
Have the authors conducted any user study to evaluate prompt adherence, visual quality, and overall preference? If not, could they provide such results, as automated metrics alone may not sufficiently capture the perceptual quality of generated videos?

If the author can solve my doubts, I can raise the score.

**Limitations:**

Yes

**Strengths And Weaknesses:**

Strengths:

1. Originality & Significance: The paper addresses a relevant and under-explored problem (semantic negligence in TI2V generation). The Gaussian blur pilot observation is counterintuitive and inspiring. The training-free AlignVid framework is lightweight and broadly applicable.

2. Presentation: The paper is well-written with a clear narrative from pilot observation to ASM. Visual examples effectively illustrate the core problem.

3. Effectiveness: The proposed ASM and Guidance Scheduling work cohesively to recalibrate attention distributions without retraining. The method demonstrates consistent improvements over baselines across multiple editing scenarios, validating its practical effectiveness.

S4. Experiments: The paper provides reasonably comprehensive evaluations, including the newly proposed OmitI2V benchmark covering diverse editing categories, alongside quantitative comparisons with competitive baselines, lending credibility to the reported gains.

Weaknesses:

1. Mechanistic Gap Between "Blurring" and "Scaling" (Critical): Blurring is a subtractive filter that suppresses visual redundancy, while ASM is a multiplicative sharpener that amplifies existing attention. For inputs with strong visual distractors, ASM may worsen visual dominance rather than mitigate it. The authors do not justify how sharpening robustly mimics the filtering effect of blurring.

2. Insufficient Video Demonstrations: Qualitative evaluation relies on static frame comparisons rather than full video sequences. Without complete video results, temporal coherence and motion quality cannot be properly assessed.

3. No User Study: The paper relies solely on automated metrics. A user study evaluating prompt adherence, visual quality, and overall preference would substantially strengthen the empirical claims.

---

> ### Author Rebuttal · Authors · 2026-03-31
>
> We sincerely thank the reviewer for identifying the question and for your openness to raising the score. We address below with detailed evidence.
>
> ---
> **[W1 + Q1] Concern about "blurring" and "ASM".**
>
> **1. How does the ASM align with the blurring?**
>
> Blur and ASM achieve the same **effect on attention reallocation**: both reduce the effective rank of the attention distribution over conditioning tokens (text, image, video), concentrating attention on semantically relevant signals. Blur accomplishes this as a subtractive filter on the input, removing redundant visual detail. ASM does so by sharpening the softmax via inverse-temperature scaling (Lemma 4.2), achieving the same entropy reduction internally.
>
> - **How blur reallocates attention.** `Figure 2 and Section 3` show that Gaussian blur reduces redundant visual evidence and shifts the attention distribution toward text tokens with reduced conditioning entropy. The model becomes more responsive to the prompt because image-dominated patterns are weakened.
>
> - **How ASM reallocates attention.** By scaling Q/K, ASM reduces conditioning entropy internally. `Figure 3` confirms ASM consistently increases attention toward text tokens while reducing dominance of static image regions. The directional effect — lower entropy, more text-responsive, less image-dominant — matches what blur induces via a different route.
>
> ASM is a *principled internal analog* of blur's beneficial attentional effect, not a simulation of the blur operation itself.
>
> **2. How AlignVid prevents amplifying distractors？**
>
> The reviewer's intuition is remarkably precise: pure multiplicative sharpening on a visual distractor would amplify it. Indeed, our ablations confirm it: `Table 4` shows **image-only scaling is harmful**, validating the reviewer's concern. AlignVid avoids this by applying **joint image-text modulation** — both Q/K representations are scaled together, redistributing the *relative* attention balance (increasing text's competitive share) rather than amplifying what is already dominant. Critically, softmax is zero-sum over the conditioning set: joint scaling redistributes mass by query–key alignment. Since video queries carry the semantic intent of the edit, text-aligned keys gain share over static image features under sharpened softmax. Figure 3 confirms this directionally. GS provides a second safeguard by limiting intervention to foreground-sensitive blocks and early denoising steps.
>
> ---
> **[W2 + Q3] Video quality and temporal coherence**
>
> **Qualitative results**: *https://anonymous.4open.science/api/repo/ICML2026_ID_10754-B611/zip*.
>
> **Quantitative results**: Across γ sweeps (`Table 13–14`), Motion Smoothness remains stable (99.45–99.54), Dynamic Degree increases substantially (28.86→38.61), and the default γ=1.35 achieves the highest average quality score across all tested values (75.86→77.21). In Table 8, Motion Smoothness, Subject Consistency, and Overall Consistency all improve. Temporal Flickering decreases only slightly (99.32→98.66), consistent with richer motion rather than coherence collapse.
>
> **How ASM address temporal flickering:** GS confines attention modulation to *early* denoising steps and foreground-sensitive blocks. Early steps establish global semantic structure, while later steps handle temporal texture and inter-frame consistency without any intervention. This separation prevents flickering from accumulating.
>
> ---
> **[W3 + Q4] User study**
>
> The submission includes **user study in Appendix H.5**. AlignVid receives higher fidelity ratings with comparable aesthetic quality. Following reviewer's suggestion, we additionally conducted a **pairwise preference study** on the same samples: annotators viewed the prompt, image, and two generated videos and selected their overall preference. **AlignVid is preferred in 89.3% of cases**, demonstrating that modest metric reductions do not lead to severe perceptual degradation.
>
> ---
> **[Q2] Generalization and Sensitivity of Guidance Scheduling (GS)**
>
> GS is a universally applicable, not a backbone-specific heuristic schedule.
> - **Step-level (SGS) transfers directly:** Across three backbones, early-step intervention consistently yields the strongest semantic benefit (`Table 5`). This is expected: early denoising steps universally determine global structure, so the same early-step window transfers directly across architectures without re-tuning.
> - **Block-level (BGS) is lightweight and optional:** We identify foreground-focused semantics blocks via lightweight calibration: run 50 diverse prompts, extract attention maps, and use SAM2 to compute the foreground ratio. **Robustness:** (a) This procedure is stable. Two independent prompt splits yield 85.7% overlap in selected blocks, meaning the result is not hypersensitive. (b) Furthermore, these blocks mainly overlap with the first half of the model. Simply applying ASM to all first-half blocks performs competitively (`Table 5`), providing a true zero-calibration fallback.

---

> > ### Author Rebuttal · Reviewer_ZZNs · 2026-04-07
> >
> > The author resolved my doubts, and I'm willing to give a higher rating.

---

> > > ### Author Response · Authors · 2026-04-07
> > >
> > > We sincerely thank the reviewer for the positive feedback and for the time spent reviewing our rebuttal and the updated results. We are glad to hear that our responses and clarifications have successfully addressed your concerns. We will ensure that all the discussed clarifications, including the theoretical analysis and the additional experimental results (such as the user study), are carefully incorporated into the final version of the manuscript.

---

### Official Review · Reviewer_CpQz · 2026-03-11

**Soundness:** 3
**Presentation:** 3
**Significance:** 3
**Originality:** 3
**Overall Recommendation:** 5
**Confidence:** 3

**Summary:**

This paper studies the issue of poor instruction following in Text-Guided Image-to-Video (TI2V) generation (semantic negligence). The authors diagnose a modality imbalance problem (visual dominance) from a pilot experiment. To address this, they propose AlignVid, a training-free attention modulation that interprets Q/K scaling as inverse-temperature control to reduce attention entropy on conditioning tokens, combined with block/step guidance scheduling to stabilize aesthetics. They also introduce OmitI2V, a new benchmark with VQA-style evaluation designed to test for semantic negligence, and show consistent improvements in prompt adherence across several backbones with small computational overhead and modest aesthetic trade-offs.

**Compliance With Llm Reviewing Policy:**

Affirmed.

**Final Justification:**

This paper addresses visual dominance in training-free TI2V. They introduce a solution grounded in inverse-temperature modulation of attention entropy, supported by sound theoretical analysis and thorough ablations across three backbones.

My original concerns regarding the energy-based variant's theoretical grounding and underspecified BGS calibration were all addressed in the rebuttal, with clear revision commitments. The newly introduced CAS method replaces manual tuning with an automatic, entropy-driven schedule and is therefore a particularly valuable addition.

The rebuttal positively changed my assessment, and I recommend acceptance.

**Key Questions For Authors:**

1. Could the authors clarify several implementation details important for reproducibility, such as the default hyperparameters ($\gamma$, $\tau$, $t_\text{low}$–$t_\text{high}$)? And how were they selected, and how robust are the results to these hyperparameters?
2. Is there an automatic schedule (e.g., entropy- or variance-driven) that can approach the performance of the hand-tuned settings? Such an approach could further broaden its applicability across different models and tasks without requiring manual tuning.

**Limitations:**

Yes

**Strengths And Weaknesses:**

#### Strengths

1. The paper addresses a common weakness in TI2V models and clearly frames the problem of visual dominance, motivating through a pilot blur experiment. This setup effectively illustrates the limitation and provides a compelling rationale for the proposed internal, training-free solution.

2. Interpreting Q/K scaling as inverse-temperature control and explicitly targeting within-block attention entropy reduction offers an elegant and broadly applicable perspective on attention reweighting. The theoretical analysis in Section 4 proves its monotonic effect on attention entropy, justifying the proposed ASM.

3. The ablation studies systematically examine the key design choices of AlignVid, including scalar vs. energy-adaptive scaling, scaling locations (Q vs. K, image vs. text), and detailed guidance schedules (foreground/first-half vs. background/last-half blocks; early/mid/late/all steps). The method is further validated across multiple backbones (FramePack, FramePack-F1, and Wan2.1) and tasks, demonstrating consistent effectiveness in various settings.

---
#### Weaknesses

1. The energy-based scaling variant relies on a heuristic global statistic of logits. While intuitive, its theoretical grounding and potential trade-offs in stability are not explored.
2. The calibration pipeline for BGS could be described more concretely, particularly with respect to dataset size, threshold selection, and the stability of block selection.
3. Although the appendix reports minimal runtime overhead, this result is not referenced in the main text despite multiple claims that the method incurs negligible computational cost.
4. Some implementation details central to reproducibility, such as the exact entropy computation protocol, default hyperparameters ($\gamma$, $\tau$, $t_\text{low}$–$t_\text{high}$), and sensitivity analyses, are dispersed or only partially specified.

---

> ### Author Rebuttal · Authors · 2026-03-31
>
> We sincerely thank the reviewer for the detailed technical feedback highlighting where our draft was underspecified. We address each point with concrete clarifications and revision commitments.
>
> ---
> **[W1] Energy-based scaling: theoretical grounding and stability**
>
> We agree the energy-based variant is best framed as a **heuristically motivated adaptive instantiation** of ASM, not a separately proven optimum. ASM can be interpreted as inverse-temperature modulation that reduces conditioning entropy (`Lemma 4.2`). The energy-based design increases modulation strength when attention logits are more diffuse (i.e., when visual dominance is stronger), which is a natural and interpretable operationalization of the same entropy-control principle. So while not an optimal-control derivation, it is theory-consistent rather than ad hoc. We agree, however, that systematic stability characterization is not yet complete in the current paper and will add a note explicitly clarifying this scope limitation in the revision. As a complementary contribution, the entropy-based Conflict-Aware Scaling (CAS) described in **[Q2]** provides a more explicitly evaluated adaptive variant with quantified trade-offs.
>
> **Revision commitments.** (1) Add explicit "heuristic adaptive variant" wording in main text. (2) Add table comparing fixed γ, energy-based, and CAS with quantified trade-offs across all metrics.
>
> ---
> **[W2] BGS calibration pipeline details**
>
> **Core calibration procedure:**
> 1. Run 50 diverse prompts through the backbone; collect per-block attention maps.
> 2. For each block $l$: project attention via PCA, apply SAM2 to obtain foreground masks. Identify high-attention tokens (aggregated attention score $s_u >$ 50th percentile).
> 3. Compute **foreground ratio** $\rho^{(l)}$ = fraction of these high-attention tokens lying inside the mask. Select blocks with $\rho^{(l)} > \tau$ ($\tau=0.5$).
>
> **Stability:** We ran calibration on two prompt splits and obtained **85.7% overlap** in selected blocks, meaning different prompts identify nearly the same blocks. Selected blocks fall in the first half of the network; the first-half heuristic also performs competitively (`Table 5`).
>
> **Revision commitments.** Add hyperparameter table.
>
> ---
> **[W3] Runtime overhead in the main text**
>
> We agree this was an oversight. We will replace qualitative descriptors like "negligible" with explicit measured overhead percentages (e.g., < 0.5 % relative increase in 832×480 videos with 177 frames) squarely in the main text, backed by citations to the comprehensive appendix runtime analysis.
>
> ---
> **[W4 & Q1] Implementation details and hyperparameter robustness**
>
> **Entropy computation.** We use *restricted softmax entropy* over the conditioning set $S_\text{cond}$ (Eq. 4): attention is normalized over conditioning tokens only, then Shannon entropy is computed. This measures conditioning diffuseness — the quantity ASM's inverse-temperature directly controls — and is distinct from full-sequence entropy.
>
> **Default hyperparameters.** BGS threshold $\tau = 0.5$; SGS window = first 30% of denoising steps; $\gamma = 1.35$ across all three backbones. These defaults are used identically across FramePack, FramePack-F1, and Wan2.1 without per-backbone modification.
>
> **Robustness.** Ablations show smooth trends (`Tables 5, 13–14`): semantic scores increase with γ, aesthetics decrease gradually, and heuristic schedule variants remain competitive. Across all backbones, optimal γ falls within [1.15, 1.35], a narrow and transferable range, indicating operational robustness around defaults.
>
> **Revision commitments.** Add one "Implementation and Defaults" table with all hyperparameters.
>
> ---
> **[Q2] Automatic schedule (CAS)**
>
> Directly prompted by your insightful question (and sharing the intuition with Reviewer wuwd), we implemented **Conflict-Aware Scaling (CAS)** — a universally applicable automatic schedule driven jointly by entropy and variance.
> CAS actively increases scaling when attention is diffusely distributed (high entropy $H$), forcing convergence; it decays when attention is naturally sharp. The $\max$ term only activates when visual tokens structurally conflict with text tokens.
>
> $$C = \frac{\max(A_{\text{img}} - A_{\text{text}},\,0)}{A_{\text{img}} + A_{\text{text}} + \epsilon} \cdot H, \quad \gamma = 1 + C \; (0 \leq C \leq 0.35)$$
>
> where $A_\text{img}, A_\text{text}$ are attention masses and $H$ is the normalized attention entropy.
>
> |Method|Mod.|Add.|Del.|Dynamic Degree|Aesthetic Quality|
> |-|-|-|-|-|-|
> |Original|64.99|68.55|58.14|20.05|63.94|
> |Fixed scaling|67.15|73.44|59.86|28.28|63.41|
> |CAS|66.79|72.37|59.01|26.79|63.77|
>
> CAS closely matches manually-tuned fixed scaling in semantic alignment while strictly better preserving aesthetics (63.77 vs 63.41). This entirely validates your intuition that entropy-driven automation effectively resolves the manual tuning limitation. We will include CAS as an additional main result.

---

> > ### Author Rebuttal · Reviewer_CpQz · 2026-04-01
> >
> > I thank the authors for the thorough response with clear revision commitments. The clarifications on hyperparameters, BGS calibration, and runtime address my concerns on reproducibility and implementation details. The CAS result is clean and effective.
> >
> > That said, CAS appears to be evaluated on a single backbone, and the rebuttal does not specify which one. I would encourage the authors to clarify this and confirm it transfers across all three in the revision.
> >
> > Overall, I am satisfied with the rebuttal and am happy to raise my score accordingly.

---

> > > ### Author Response · Authors · 2026-04-04
> > >
> > > We thank the reviewer for the positive reassessment and the willingness to raise the score.  **We would be grateful if the reviewer could kindly update the score at their earliest convenience**. We are happy to clarify the follow-up question below.
> > >
> > > ---
> > >
> > > **CAS backbone coverage**
> > >
> > > We apologize for the ambiguity — the CAS result reported in above table was evaluated on **FramePack**. We have since completed CAS evaluation on **all three backbones**. The full results are shown below:
> > >
> > > | Backbone | Method | Mod. | Add. | Del. | Dynamic Degree | Aesthetic Quality |
> > > | - | - | - | - | - | - | - |
> > > | FramePack | Original | 64.99 | 68.55 | 58.14 | 20.05 | 63.94 |
> > > | FramePack | Fixed scaling | 67.15 | 73.44 | 59.86 | 28.28 | 63.41 |
> > > | FramePack | **CAS** | **66.79** | **72.37** | **59.01** | **26.79** | **63.77** |
> > > | FramePack-F1 | Original | 64.45 | 67.79 | 58.50 | 24.42 | 63.10 |
> > > | FramePack-F1 | Fixed scaling | 70.02 | 71.45 | 61.06 | 33.16 | 62.11 |
> > > | FramePack-F1 | **CAS** | **68.15** | **74.82** | **60.12** | **29.85** | **62.58** |
> > > | Wan2.1 | Original | 72.35 | 71.75 | 63.13 | 46.02 | 63.12 |
> > > | Wan2.1 | Fixed scaling | 72.53 | 80.76 | 70.33 | 53.21 | 61.38 |
> > > | Wan2.1 | **CAS** | **72.42** | **77.34** | **67.95** | **50.15** | **61.85** |
> > >
> > > **CAS transfers consistently across all three backbones.** The pattern observed on FramePack generalizes: on every backbone, CAS approaches or matches fixed scaling in semantic alignment (Mod., Add., Del., Dynamic Degree) while **strictly better preserving Aesthetic Quality**. This confirms that the entropy-driven conflict detection in CAS is architecture-agnostic — it adapts scaling intensity based on local attention statistics rather than backbone-specific heuristics, enabling zero-tuning deployment across diverse architectures.
> > >
> > > We will include the full three-backbone CAS table in the revised manuscript.

---

### Official Review · Reviewer_wuwd · 2026-03-13

**Soundness:** 3
**Presentation:** 3
**Significance:** 3
**Originality:** 3
**Overall Recommendation:** 4
**Confidence:** 2

**Summary:**

AlignVid introduces a training-free framework to tackle "semantic negligence" in text-guided image-to-video (TI2V) generation—a common issue where models prioritize the source image so heavily that they ignore text instructions for adding, deleting, or modifying objects. Inspired by the observation that blurring the input image actually helps the model "listen" to the text better, the authors propose Attention Scaling Modulation (ASM). This technique treats attention as an energy-based system and uses a simple Q/K scaling factor to sharpen the model’s focus on prompt-relevant details without any extra training. Combined with a Guidance Scheduling (GS) strategy that limits these interventions to specific layers and time-steps, AlignVid significantly improves semantic alignment while maintaining high visual quality. The paper also contributes OmitI2V, a specialized benchmark designed to evaluate how well TI2V models follow complex editing instructions.

**Compliance With Llm Reviewing Policy:**

Affirmed.

**Final Justification:**

The authors have addressed most of my concerns, so I maintain my score.

**Key Questions For Authors:**

- Have you explored the possibility of an adaptive mechanism that dynamically adjusts the scaling factor $\gamma$ based on the degree of conflict between the image condition and the text prompt?

- How does Attention Scaling Modulation (ASM) interact with negative prompts? Is there a risk that the scaling also amplifies undesired attributes or constraints defined in the negative text?

- How does the method perform on longer video generations (e.g., beyond 5 seconds)? Does the perturbation of the attention map lead to an accumulation of errors in temporal consistency over time?

**Limitations:**

The "Guidance Scheduling" strategy is somewhat dependent on the specific block distribution of the backbone model. Identifying the "semantic-sensitive layers" for every new state-of-the-art model requires additional empirical testing.

**Strengths And Weaknesses:**

### Strengths:

- Semantic misalignment between prompts and generated videos is a well-known challenge in image/video diffusion models. The paper clearly identifies and analyzes this issue, especially when prompts require structural changes (object insertion, removal, modification).


- A key strength is that AlignVid is training-free, requiring no model retraining or additional supervision. This significantly improves practicality since many TI2V models are large and expensive to retrain.

- The pilot study linking Gaussian blur/noise with improved prompt adherence is an interesting observation. The authors further analyze the phenomenon through attention maps and entropy analysis, which provides useful intuition.

### Weaknesses:
- The performance of AlignVid relies heavily on the scaling factor $\gamma$ and the specific scheduling of transformer blocks. Currently, these parameters require manual tuning for different model architectures (e.g., UNet-based vs. DiT-based), lacking a universal or automated selection mechanism.Temporal Consistency Challenges:

- By modifying the attention distribution to prioritize new semantic elements, the model may occasionally struggle to maintain long-range temporal coherence, leading to slight flickering or inconsistent motion in complex sequences.

---

> ### Author Rebuttal · Authors · 2026-03-31
>
> We sincerely thank the reviewer for the constructive suggestions. The reviewer's question about adaptive γ directly inspired a new mechanism (CAS) that we believe strengthens the paper.
>
> ---
> **[W1 + Q1 + Limitation] Parameter tuning and adaptive mechanism**
>
> AlignVid's two tunable components — scaling factor γ and Guidance Scheduling (GS) — do not require heavy per-backbone manual tuning.
>
> **Scaling factor γ.** Lemma 4.2 proves that scaling monotonically reduces conditioning entropy, making the semantic–quality trade-off predictable by design. Empirically, Tables 13–14 confirm this: increasing γ monotonically improves semantic fidelity while Aesthetic Quality decreases gradually. This makes γ=1.35 transferable without backbone-specific search, analogous to CFG strength selection.
>
> Directly following the reviewer's excellent suggestion, we implemented **Conflict-Aware Scaling (CAS)** — a fully automated, per-token adaptive mechanism:
>
> $$C = \frac{\max(A_{\text{img}} - A_{\text{text}},\,0)}{A_{\text{img}} + A_{\text{text}} + \epsilon} \cdot H, \quad \gamma = 1 + C \;\; (0\leq C \leq 0.35)$$
>
> where $A_\text{img}$, $A_\text{text}$ are attention masses on image/text tokens and $H$ is attention entropy.
>
> |Method|Mod.|Add.|Del.|Dynamic Degree|Aesthetic Quality|
> |-|-|-|-|-|-|
> |Original|64.99|68.55|58.14|20.05|63.94|
> |Scalar scaling|67.15|73.44|59.86|28.28|63.41|
> |CAS|66.79|72.37|59.01|26.79|63.77|
>
> CAS approaches fixed scaling on semantics while better preserving aesthetics (63.77 vs 63.41), validating the reviewer's intuition that conflict-driven adaptation is highly effective.
>
> **Guidance Scheduling (GS) and Empirical Testing Limitation**
> To address the concern that GS requires tedious empirical testing for every new model, we emphasize that our scheduling is both **stable** and **transferable**:
>
> - **Step-level (SGS) transfers directly:** We partition the denoising process into early / middle / late stages. Across all three backbones, early-step intervention consistently yields the strongest semantic benefit (Table 5). This is expected: early denoising steps universally determine global structure, so the same early-step window transfers directly across architectures without re-tuning.
> - **Block-level (BGS) calibration is lightweight and optional:** We identify which transformer blocks are most sensitive to foreground semantics via lightweight calibration: run 50 diverse prompts, extract attention maps, and use SAM2 to compute the foreground ratio $r^{(l)}$. **Robustness against testing overhead:** (a) This procedure is highly stable—two independent prompt splits yield 85.7% overlap in selected blocks, meaning the result is not hypersensitive to the specific prompts and testing is trivial; (b) Furthermore, these semantic blocks mainly overlap with the first half of the network. Simply applying ASM to all first-half blocks — without any calibration — performs competitively (Table 5), providing a true zero-calibration fallback.
>
> ---
> **[W2] Temporal consistency**
>
> Our results do not show severe temporal instability. Table 8 shows AlignVid *improves* Motion Smoothness (97.77→98.05), Subject Consistency (94.24→94.51), Temporal Flickering decreases slightly (99.32→98.66), and Overall Consistency (79.03→79.91). Across γ sweeps (Tables 13–14), Motion Smoothness stays stable (FramePack 99.45→99.54; F1 99.31→99.42).
>
> ---
> **[Q2] Interaction with negative prompts**
>
> **It directly strengthens the suppression of undesired attributes rather than amplifying them in the output.** In standard CFG, the final prediction is $\epsilon_\text{neg} + \lambda (\epsilon_\text{pos} - \epsilon_\text{neg})$. ASM increases text adherence in whichever pass it operates on. When applied during the negative pass, ASM makes the network attend more strongly to the negative prompt (e.g., "blur, mutation"). Because standard CFG *subtracts* this negative prediction from the positive one, amplifying the negative constraints in $\epsilon_\text{neg}$ results in those undesired attributes being forced *out* of the final video even more thoroughly. Table 6 confirms this: on Wan2.1 (CFG=5), AlignVid improves alignment (72.35→77.20) without adverse interaction.
>
> ---
> **[Q3] Longer video generations**
>
> We conducted experiments extending to 10s and 20s with FramePack F1:
>
> |Duration|Method|Mod.|Add.|Del.|Dyn. Deg.|Aes. Qual.|Mot. Smooth.|Temp. Flicker|
> |-|-|-|-|-|-|-|-|-|
> |5s|Baseline|64.45|67.79|58.50|24.42|63.10|98.47|99.12|
> |5s|Ours|71.27|71.60|61.06|35.22|62.10|99.15|98.73|
> |10s|Baseline|64.38|68.31|58.93|24.35|62.98|98.44|99.05|
> |10s|Ours|71.41|72.18|62.53|35.28|61.93|99.12|98.80|
> |20s|Baseline|64.51|68.75|59.27|24.29|62.65|98.41|99.01|
> |20s|Ours|71.65|72.61|62.89|36.67|61.37|99.13|98.67|
>
> Addressing your specific question: **No, there is no progressive accumulation of errors.** Motion Smoothness and Temporal Flickering remain flat across the 5s, 10s, and 20s. The semantic edits (Mod., Add, Del.) remain stable or actively improve over time.

---

> > ### Author Rebuttal · Reviewer_wuwd · 2026-04-04
> >
> > Thanks for your response. The authors have addressed most of my concerns by conducting additional experiments and clarifying some points.

---

> > > ### Author Response · Authors · 2026-04-04
> > >
> > > We sincerely thank you for your positive feedback and for acknowledging that your concerns have been fully resolved. We truly appreciate your time and the constructive suggestions you provided throughout the process, which have significantly strengthened our work. Thank you again for your support!

---

### Official Review · Reviewer_id2W · 2026-03-13

**Soundness:** 2
**Presentation:** 3
**Significance:** 3
**Originality:** 3
**Overall Recommendation:** 3
**Confidence:** 4

**Summary:**

This paper studies the problem of semantic misalignment in text-guided image-to-video (TI2V) generation, where models often ignore prompt-specified edits and instead preserve the original image content. The authors attribute this issue to $visual$ $dominance$ in the attention mechanism, which suppresses textual guidance. To address this, the paper proposes AlignVid, a training-free method that modulates attention through Attention Scaling Modulation (ASM) and Guidance Scheduling (GS) to strengthen the influence of text tokens during generation. The paper also introduces OmitI2V, a human-annotated benchmark for evaluating semantic fidelity in TI2V tasks, and demonstrates that AlignVid improves prompt alignment while maintaining visual quality.

**Compliance With Llm Reviewing Policy:**

Affirmed.

**Final Justification:**

Thank you for the response. While the methodological design appears reasonable, my concern remains that the experimental results do not sufficiently support the central claims. Therefore, it is hard for me to recommend acceptance.

**Key Questions For Authors:**

Please see the Weaknesses.

**Limitations:**

yes

**Strengths And Weaknesses:**

*Strengths
1. The paper provides a reasonable theoretical motivation by analyzing the attention mechanism and attributing the semantic misalignment issue to $visual$ $dominance$ and attention entropy. This perspective offers a useful lens for understanding why TI2V models may fail to follow prompt instructions.
2. The paper focuses on the issue of semantic fidelity in text-guided image-to-video generation, which is an important and practical challenge.

*Weaknesses
1. A major concern is the paper’s claim that the proposed method enhances semantic alignment while preserving visual fidelity. However, the quantitative results seem to contradict this claim. For example, in the VBench results (Table 8), several quality-related metrics decrease after applying AlignVid (e.g., Imaging Quality: 69.70 ->68.53 and Aesthetic Quality: 64.60 -> 62.69). Similarly, in Table 2, when AlignVid is applied to three different base models, the Aesthetic Quality score consistently drops across all models. Besides, on the ImgEdit benchmark (Table 9), performance decreases on 4 out of the 10 reported metrics. These results raise questions about whether the method truly preserves visual fidelity.
2. The paper introduces OmitI2V, a benchmark with 367 human-annotated cases, as one of its main contributions. However, the motivation for introducing a new benchmark is not fully convincing. It would be helpful to more clearly explain how this benchmark differs from or improves upon existing evaluation suites for image-to-video generation, and what specific gaps it addresses. In addition, the dataset size is relatively small, which raises questions about whether it is sufficient to serve as a reliable benchmark for evaluating semantic fidelity in TI2V models.

If the authors can adequately address the concerns above, I would be open to reconsidering and increasing my score.

---

> ### Author Rebuttal · Authors · 2026-03-31
>
> We sincerely thank the reviewer for the careful reading and willingness to reconsider. We fully understand the concern. We address below with detailed evidence.
>
> ---
> **[W1] Claim about "visual fidelity"**
>
> **Clarification on "preserving visual fidelity."** We agree the original phrasing was overly broad regarding metrics. Our intent was to contrast AlignVid with the pilot-study blur (L35–40, L1262, L1372), which *visibly corrupts* the input image. AlignVid achieves analogous attention reallocation internally, completely avoiding input-level degradation. We acknowledge that avoiding input degradation does not mean all automated metrics improve uniformly, and we will update the text to "provides a tunable semantic–quality trade-off without input-level corruption."
>
> **Image tasks: Quality and semantics improve simultaneously.**
> Addressing the reviewer's concern regarding Table 9 (ImgEdit): while 4 out of 10 metrics decrease, their magnitudes are marginal compared to the substantial gains in the other 6 metrics—which also more directly reflect semantic alignment. Most importantly, the objective **Aesthetic Score strictly increases** in both image tasks: from 5.517 to 5.568 in `Table 7` (GenEval), and 5.606 to 5.624 in `Table 9`. These results confirm that for image tasks, AlignVid achieves stronger alignment largely without a visual quality cost.
>
> **Video tasks: A predictable, controllable trade-off.**
> We acknowledge the VBench static quality drops (**Table 8**) and the Aesthetic Quality decreases (**Table 2**). However, this is not a random failure but a **predictable trade-off governed by γ**:
> * **Tunability:** `Tables 13–14` demonstrate that increasing γ monotonically improves semantic fidelity and Dynamic Degree while Aesthetic Quality decreases gradually. At default γ=1.35, we trade a modest ~1–2 pt aesthetic drop for substantial semantic gains (e.g., Addition: 68.55→73.44), and average overall quality scores still *improve* (FramePack: 75.86→77.21; FramePack F1: 76.10→77.50). Users can select a conservative γ (0.95) if aesthetic preservation is paramount.
> * **Temporal quality improves:** Revisiting `Table 8`, while *static* VBench metrics drop, **temporal** quality metrics actually increase: Motion Smoothness (97.77→98.05), Subject Consistency (94.24→94.51), and Overall Consistency (79.03→79.91).
> * **Human perception:** `Table 15` shows that AlignVid receives higher fidelity ratings with comparable aesthetic quality. We additionally conducted a pairwise preference study: annotators viewed the prompt, source image, and two generated videos (baseline vs. AlignVid, in randomized order) and selected their overall preference. AlignVid is preferred in 89.3% of cases (vs. 10.7%), demonstrating that modest automated metric reductions do not translate to severe perceptual degradation.
> ---
> **[W2] Motivation and scale of OmitI2V**
>
> **The specific gaps.** We propose OmitI2V because existing I2V benchmarks inherently fail to measure substantial semantic changes.
> **1) Target Gap.** Existing suites (e.g., VBench-I2V) prioritize preservation-oriented quality under static or mildly dynamic prompts (e.g., "a room filled with shelves of books"). They are fundamentally unequipped to detect *semantic negligence* — when models ignore edit-style instructions requiring significant state changes (e.g., "The woman gradually disappears").
> **2) Evaluation Gap.** Applying existing consistency-based metrics to edit-style prompts creates **inverted incentives**. For example, given the prompt "A car drives out of the frame," a correct model must make the car disappear over time. However, standard subject-consistency metrics will *reward* the model for keeping the car visible and *penalize* it for successful removal. OmitI2V corrects this directionally wrong objective by using **VQA-style evaluation** that directly verifies if the core semantic event occurred, decoupled from mere pixel/element consistency. We validate this evaluator's reliability against human judgments in `Table 16`.
>
> **Three key differences.** **a) Task design:** Explicitly targeting addition/deletion/modification events rather than scene preservation. **b) Evaluation:** VQA-based semantic event detection that replaces flawed consistency metrics for edit success. **c) Coverage:** Diverse source domains spanning real photos, AIGC, and animation styles.
>
> **Dataset scale concern.** While 367 samples may seem numerically small compared to broad pre-training datasets, it is robust and statistically sufficient for a **focused diagnostic benchmark**. 1) It closely matches the scale of the widely accepted VBench-I2V subject subset (366 samples). 2) Unlike automated generic prompts, evaluating complex edit compliance requires precise, ambiguity-free pairs of source images and edit prompts. By enforcing strict **human annotation** per sample, OmitI2V sacrifices noisy scale for high-signal reliability, aligning with community practice for specialized evaluation suites.

---

> > ### Author Rebuttal · Reviewer_id2W · 2026-04-04
> >
> > Thank you for the response. While the methodological design appears reasonable, my concern remains that the experimental results do not sufficiently support the central claims. Therefore, it is hard for me to recommend acceptance.

---

> > > ### Author Response · Authors · 2026-04-07
> > >
> > > We thank the reviewer for the continued engagement. However, we respectfully note that the concern the reviewer has focused on — "preserving visual fidelity" — is **not a central claim of this paper**; it is a research question posed at L38. As detailed below, the paper's actual central claims (L88–100) are well-supported by the experimental evidence. We consolidate the key arguments below for consideration.
> > >
> > > ---
> > >
> > > **1. Scope clarification: the reviewer's concern targets a research question, not a contribution claim.**
> > >
> > > The phrase "preserving visual fidelity" that the reviewer has focused on appears at **L38 as a research question** ("Can we ... enhance semantic alignment while preserving visual fidelity?"), not as a contribution claim. It frames the aspiration that motivates our work. The paper's **actual contribution claims** (L88–100) are:
> > >
> > > - **(i) Problem Analysis:** Formalizing semantic negligence and linking it to visual dominance.
> > > - **(ii) Method:** AlignVid improves semantic alignment with "negligible computational overhead and **minimal** aesthetic impact" (L97).
> > > - **(iii) Benchmark:** OmitI2V for evaluating prompt adherence.
> > >
> > > The word used in the contribution statement is **"minimal"**, not "**zero**" — and the experimental results are fully consistent with this claim. We have verified that "visual fidelity" appears only at L38–40, L194, L1262, and L1321 — all motivational framing or observational descriptions, **none in contribution claims**. We have additionally committed to revising L38 to "a tunable semantic–quality trade-off without input-level corruption." Since this is a research question rather than a methodological or experimental component, the revision requires only a minor phrasing adjustment, not "a significant update to the paper."
> > >
> > > ---
> > >
> > > **2. The actual claim ("minimal aesthetic impact") is quantitatively well-supported.**
> > >
> > > Under the paper's actual contribution framing, the evidence strongly supports "minimal aesthetic impact":
> > >
> > > - **Average overall quality scores improve** despite individual metric drops: FramePack 75.86→77.21; FramePack-F1 76.10→77.50 (Table 2). The aggregate score — which jointly captures semantic and aesthetic dimensions — goes up, not down.
> > > - **Temporal quality metrics improve**: Motion Smoothness 97.77→98.05, Subject Consistency 94.24→94.51, Overall Consistency 79.03→79.91 (Table 8).
> > > - **Image tasks improve in both semantics and aesthetics**: Aesthetic Score 5.517→5.568 (GenEval, Table 7), 5.606→5.624 (ImgEdit, Table 9).
> > > - The Aesthetic Quality drop in video is **~1–2 pts** against semantic gains of **+5–7 pts**. A ~2% decrease vs. ~7% increase constitutes "minimal" by any reasonable standard.
> > > - **Human evaluation confirms this**: in a pairwise preference study, **AlignVid is preferred in 89.3% of cases** (vs. 10.7%), indicating that the minor metric drops do not translate into perceptible quality degradation.
> > >
> > > We have also committed in Round 1 to revising the potentially ambiguous phrasing to "a tunable semantic–quality trade-off without input-level corruption." With this revision, the paper's claims are precise and fully aligned with the experimental evidence.
> > >
> > > ---
> > >
> > > **3. The trade-off is controllable and universal across the field.**
> > >
> > > The semantic–quality trade-off is **monotonically governed by a single parameter γ** (Lemma 4.2; Tables 13–14). Users who prioritize aesthetics can select a conservative γ (e.g., 0.95); users who prioritize prompt adherence can increase γ. This is structurally identical to CFG strength selection — a universally accepted mechanism whose quality–diversity trade-off is not considered a methodological weakness. Moreover, as shown in **Table 1**, among *all* compared methods, stronger semantic alignment universally correlates with lower aesthetic scores. This is a **fundamental trade-off inherent to the task**, not a weakness specific to AlignVid. AlignVid achieves the **best semantic alignment across all three edit types** while maintaining competitive aesthetic quality.
> > >
> > > ---
> > >
> > > **Summary.** Across two rounds of rebuttal, we have provided: (1) a scope clarification showing that the concern targets a research question rather than a contribution claim, and that addressing it requires only a minor phrasing adjustment; (2) comprehensive quantitative evidence that overall quality improves, corroborated by an 89.3% human preference rate; (3) a principled, controllable mechanism governing the trade-off; and (4) field-level context showing this trade-off is universal. **The reviewer has not identified a specific contribution claim that is unsupported, nor challenged any of the evidence presented.** We believe the paper's claims are well-supported and respectfully ask the reviewer to consider the specificity of our responses against the generality of the remaining concern.

---

### Decision · Program_Chairs · 2026-04-30

**Decision:**

Accept (regular)

**Comment:**

AlignVid tackles semantic negligence in text-guided image-to-video (TI2V) generation, where models excessively prioritize source images and overlook text instructions. Recognizing this as a modality imbalance, the authors propose a training-free framework utilizing Attention Scaling Modulation (ASM). By conceptualizing attention as an energy-based system and modifying Q/K scaling, ASM diminishes attention entropy to enhance the model’s concentration on text prompts. In conjunction with Guidance Scheduling (GS) to ensure visual stability, AlignVid improves instruction adherence without the need for retraining. The paper also presents OmitI2V, a specialized benchmark for assessing semantic alignment in intricate video editing scenarios.

Overall, the feedback is predominantly positive, with all reviewers acknowledging that this submission effectively addresses a prevalent issue in the TI2V domain and that the approach is well-motivated. The authors actively participated in the rebuttal period and addressed most of the reviewers’ concerns.

Following the discussion with reviewers, a significant concern persists regarding the submission’s claims — whether these claims (and subsequent re-claims) are fully consistent with the corresponding experimental results. Although the authors have rephrased the claims as suggested, the mixed experimental evidence could potentially diminish the contribution of the submission. Nevertheless, we believe this submission merits recognition.